# Bubbleformer: Forecasting Boiling with Transformers

**Sheikh Md Shakeel Hassan**[1]* **Xianwei Zou**[1]
**Akash Dhruv**[2] **Aparna Chandramowlishwaran**[1]*
[1]University of California, Irvine    [2]Argonne National Laboratory

## Abstract

Modeling boiling—an inherently chaotic, multiphase process central to energy and thermal systems—remains a significant challenge for neural PDE surrogates. Existing models require future input (e.g., bubble positions) during inference because they fail to learn nucleation from past states, limiting their ability to autonomously forecast boiling dynamics. They also fail to model flow boiling velocity fields, where sharp interface–momentum coupling demands long-range and directional inductive biases.

We introduce **Bubbleformer**, a transformer-based spatiotemporal model that forecasts stable and long-range boiling dynamics including nucleation, interface evolution, and heat transfer without dependence on simulation data during inference. Bubbleformer integrates factorized axial attention, frequency-aware scaling, and conditions on thermophysical parameters to generalize across fluids, geometries, and operating conditions. To evaluate physical fidelity in chaotic systems, we propose interpretable physics-based metrics that evaluate heat flux consistency, interface geometry, and mass conservation. We also release **BubbleML 2.0**, a high-fidelity dataset that spans diverse working fluids (cryogens, refrigerants, dielectrics), boiling configurations (pool and flow boiling), flow regimes (bubbly, slug, annular), and boundary conditions. Bubbleformer sets new benchmark results in both prediction and forecasting of two-phase boiling flows.

## 1 Introduction

Boiling is one of the most efficient modes of heat transfer due to the large latent heat of vaporization at liquid-vapor interfaces, making it an attractive solution for ultra-high heat flux applications such as nuclear reactors and next-generation computing infrastructure. Companies such as ZutaCore[2] and LiquidStack[3] are pioneering two-phase and immersion cooling technologies for data centers supporting AI workloads. These industrial efforts reflect a broader trend toward harnessing phase-change phenomena for thermal control in compact, high-density environments. Boiling also holds promise for spacecraft thermal control, but without buoyancy to aid vapor removal, boiling in microgravity faces severe challenges, limiting its current viability. Addressing these limitations will require novel boiling architectures optimized for low gravity, guided by modeling, design optimization, and experimental validation–an iterative process that is computationally intensive and costly. More fundamentally, accurately modeling two-phase pool and flow boiling remains one of the grand challenges in fluid dynamics. The underlying physics is inherently chaotic and multiscale: bubbles nucleate stochastically on heated surfaces, liquid-vapor interfaces continuously deform, coalesce, and break apart, and transitions between flow regimes (e.g., bubbly, slug, annular) occur unpredictably under strong thermal-hydrodynamic coupling. As illustrated in Figure 1, boiling

---

*Corresponding authors: {sheikhh1,amowli}@uci.edu
[2]https://zutacore.com
[3]https://www.liquidstack.com

39th Conference on Neural Information Processing Systems (NeurIPS 2025) Track on Datasets and Benchmarks.

systems can span a vast range of spatial and temporal scales and are highly sensitive to boundary conditions, fluid properties, and geometry.

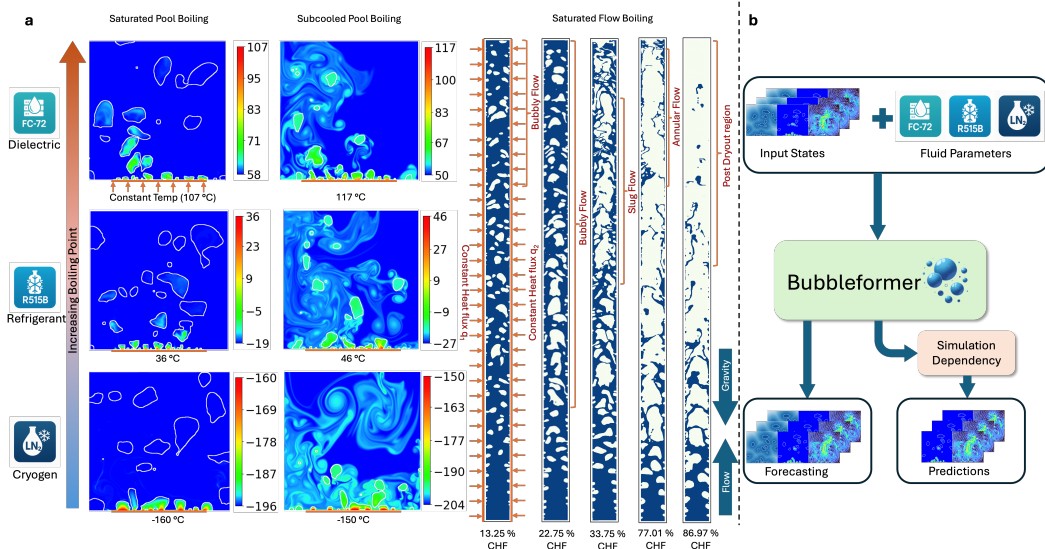

Figure 1: **(a) BubbleML 2.0 dataset**: Visualizes temperature and phase fields across pool and flow boiling configurations. Bubbles nucleate on heated surfaces, deform, coalesce, and transition between regimes, with dynamics strongly coupled to boundary conditions, working fluid, and geometry. **(b) Bubbleformer downstream tasks**: Autonomous *forecasting* of full-field dynamics including nucleation and interface evolution, and *prediction* conditioned on future bubble interfaces.

Recent advances in high-fidelity multiphysics solvers such as Flash-X [15, 10] have enabled simulations of boiling by solving incompressible Navier-Stokes equations in two phases with interface tracking and nucleation models. Although physically accurate, these simulations are computationally expensive, often requiring days on petascale supercomputers to simulate seconds of physical time [11, 12]. This has motivated the development of machine learning (ML) surrogates that learn to approximate spatiotemporal evolution of boiling directly from data. These models promise orders-of-magnitude acceleration, enabling new capabilities in real-time forecasting, parametric studies, and design-space exploration. However, current ML models for boiling [25, 32] exhibit three main limitations: (1) They require future bubble positions as input to predict velocity and temperature fields, making them unsuitable for forecasting, (2) They fail to learn bubble nucleation, a stochastic discontinuous process central to long rollouts, and (3) They fail to predict velocity fields in flow boiling, even when provided with future bubble positions.

We introduce **Bubbleformer**, a transformer-based spatiotemporal model that forecasts full-field boiling dynamics that includes temperature, velocity, and signed distance fields representing interfaces, setting a new benchmark for ML-based boiling physics. Bubbleformer makes the following core contributions:

- **Beyond prediction to forecasting.** By operating directly on full 5D spatiotemporal tensors and preserving temporal dependencies, Bubbleformer learns nucleation, key to forecasting and predicting long-range dynamics. Unlike prior models that compress time or require future bubble positions, our approach infers them end-to-end.

- **Generalizing across fluids and flow regimes.** We introduce **BubbleML 2.0**, the most comprehensive boiling dataset to date, comprising over 160 high-fidelity simulations across diverse fluids (e.g., cryogenics, refrigerants, dielectrics), boiling configurations (pool and flow), heater geometries (single- or double-sided heating), and flow regimes (bubbly, slug, and annular until dryout). Bubbleformer is conditioned on thermophysical parameters, allowing a single model to generalize across these axes.

- **Physics-based evaluation.** We introduce new interpretable metrics to assess physical fidelity beyond pixel-wise error. These include heat flux divergence, Eikonal equation loss for signed

distance functions, and conservation of vapor mass. Together, these metrics provide a more rigorous evaluation of physical correctness in chaotic boiling systems.

To our knowledge, *Bubbleformer is the first model to demonstrate autonomous, physically plausible forecasting of boiling dynamics.* It sets new state-of-the-art benchmarks on both prediction and forecasting tasks in BubbleML 2.0, representing a significant step toward practical, generalizable ML surrogates for multiphase thermal transport.

## 2 Problem Statement

Boiling involves the phase change of a liquid into vapor at a heated surface, forming bubbles that enhance turbulence and heat transfer. This phenomenon is highly chaotic: bubbles nucleate unpredictably on the heated surface, then grow, merge, and eventually detach, all while interacting with the surrounding liquid. The boiling system is governed by the incompressible Navier-Stokes equations (for momentum conservation) coupled with an energy transport equation, solved in both the liquid and vapor phases. These equations describe the evolution of the fluid's velocity $\vec{u}$ and pressure fields $P$ along with the temperature field $T$ that captures heat distribution. The *liquid-gas interface* is represented by a level-set function $\phi$ (signed distance field) that tracks the moving phase boundary. The interface $\Gamma$ is defined by $\phi = 0$, with $\phi > 0$ in the vapor region and $\phi < 0$ in the liquid. The governing equations are non-dimensionalized using characteristic scales defined in the liquid phase (e.g., capillary length for length scale and the terminal velocity for velocity scale) and key dimensionless numbers such as Reynolds, Prandtl, Froude, Peclet, and Stefan. To ensure consistency across both phases, vapor properties (e.g., density, viscosity, thermal conductivity, and specific heat) are specified relative to the liquid's properties.

Interfacial physics is modeled by enforcing the conservation of mass and energy at $\Gamma$, accounting for surface tension and latent heat. Jump conditions for velocity, pressure, and temperature across the interface are implemented using the Ghost Fluid Method (GFM), a numerical scheme that handles sharp discontinuities at the boundary without smearing them. The level-set interface $\phi$ is updated via an advection (convection) equation, which moves the interface with the local fluid velocity. Evaporation and condensation are governed by differences in interfacial heat flux between the two phases, coupling flow dynamics with thermal effects. The two-phase numerical simulation framework is implemented in Flash-X [15] and follows the formulation in [10]. For completeness, the governing equations and numerical modeling assumptions are provided in Appendix A.

## 3 Failure Modes of Neural Solvers for Boiling

Recent works [25, 32] have explored neural surrogates trained on Flash-X simulations [15, 10, 11] to model boiling dynamics. These models aim to predict future velocity, temperature, and interface evolution given past physical states. While promising, current architectures exhibit persistent failure modes that limit their ability to forecast real-world systems.

### 3.1 Simulation Dependency and Failure to Learn Nucleation

Current boiling models are trained to predict velocity and temperature fields given both past physical states and *future* bubble positions. A task in BubbleML [25] is to learn an operator: $\mathcal{G}(\phi_{prev}, \vec{u}_{prev}, T_{prev}, \phi_{next}) = [\vec{u}_{next}, T_{next}]$. Although this task is tractable during supervised training, it introduces a serious limitation during inference: the model requires access to $\phi_{next}$, which are not available without running the underlying simulation. This precludes autonomous rollouts and the model is fundamentally unsuitable for forecasting.

To eliminate this dependency, one can attempt to jointly learn bubble evolution: $\mathcal{G}(\phi_{prev}, \vec{u}_{prev}, T_{prev}) = [\phi_{next}, \vec{u}_{next}, T_{next}]$, requiring the model to predict future nucleation events purely from historical data. However, this task proves challenging for current architectures. Bubble nucleation is a discontinuous stochastic phenomena governed by microscale surface physics, contact angles, and thermal boundary layers [23, 58, 11]. In Flash-X, nucleation is modeled algorithmically: new bubbles reappear on the heater surface at nucleation sites with seed radii after specified wait time (time before a new bubble forms after the prior bubble departs). When these conditions are satisfied, re-nucleation is achieved through the union of the new signed

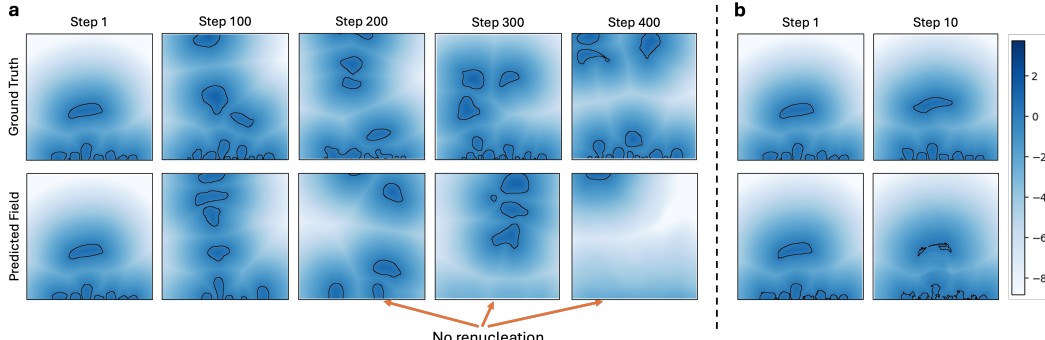

Figure 2: **Failure modes of current ML models in forecasting.** Left: UNet models maintain smooth evolution during autoregressive rollout but fail to nucleate. Right: FNO exhibits early instability.

distance field (due to the newly formed bubble) with the evolving phase field, introducing sharp discontinuities and topological changes.

These reinitializations are non-differentiable and violate assumptions of spatial smoothness. As a result, learning this behavior directly from data proves difficult. We train UNet and Fourier Neural Operator (FNO) models from BubbleML [25] to learn this task. UNet-based models perform well in single-step prediction but fail in autoregressive rollouts, as they do not learn to nucleate new bubbles as shown in Figure 2. FNO models degrade more rapidly, often diverging after a single step. FNO relies on global spectral filters and smooth continuous mappings between input and output Banach spaces[34]. The sharp discontinuities associated with nucleation add "shocks" in the underlying function space, making it difficult for the neural operator models leading to quick divergence during rollout. This is consistent with theoretical analysis showing that neural operators are sensitive to abrupt and high-frequency discontinuous topological changes[6] [35].

### 3.2  Failure in Flow Boiling Velocity Prediction

Flow boiling introduces additional modeling challenges. Unlike pool boiling, which is largely buoyancy-driven and symmetric, flow boiling features directional velocity, thin films, and shear-induced instabilities. Accurate modeling of such flows requires resolving the tight coupling between evolving interface dynamics and momentum transport (i.e., velocity fields). We observe that UNet and FNO models fail to predict velocity in flow boiling datasets. This failure occurs despite training on fine-resolution data and using future interface fields as supervision. We identify two primary contributing factors:

**Lack of directional inductive bias.** Flow boiling geometries are typically long, narrow rectangular domains (see Figure 1), with a higher resolution in the flow direction than in the cross-stream direction. When FNO's low-pass filters are applied isotropically in Fourier space, they disproportionately suppress high-frequency modes along the flow-direction, resulting in directional aliasing. Similarly, UNet models apply symmetric convolutions across both axes and lack the architectural bias to prioritize dominant features, limiting their ability to resolve streamwise velocity gradients and anisotropic flow patterns.

**Insufficient spatiotemporal integration.** Unlike pool boiling, flow boiling introduces additional spatiotemporal gradients from the bulk liquid flow. Localized changes in interface topology (e.g., film rupture, bubble coalescence) induce sharp and nonlocal responses in the velocity field [16]. Predicting this behavior requires long-range spatial context and multiscale temporal integration. UNet models, while effective for local pixelwise regression, lack sufficient temporal memory and spatial receptive field to capture these dynamics. FNOs, on the other hand, suffer from spectral oversmoothing, which makes it difficult to isolate sharp interface-driven effects.

These failures are not isolated. Across all flow boiling datasets in BubbleML, we observe that FNO and UNet models fail to converge when learning to predict velocity. This highlights the need for architectures that go beyond spectral operators and spatial attention to incorporate hierarchical, directional, and temporally-aware representations.

# 4 Bubbleformer

Bubbleformer is a spatiotemporal transformer designed to forecast boiling dynamics across fluids, boiling configurations, geometries, and flow regimes. In contrast to prior surrogates that fail to nucleate (Section 3.1) or generalize to flow boiling (Section 3.2), Bubbleformer integrates structural innovations that enable long rollout, parameter generalization, and high-frequency prediction.

## 4.1 Model Architecture

Bubbleformer architecture illustrated in Figure 3 consists of four components: hierarchical patch embedding, physics-based parameter conditioning, factorized space-time axial attention, and frequency-aware feature modulation.

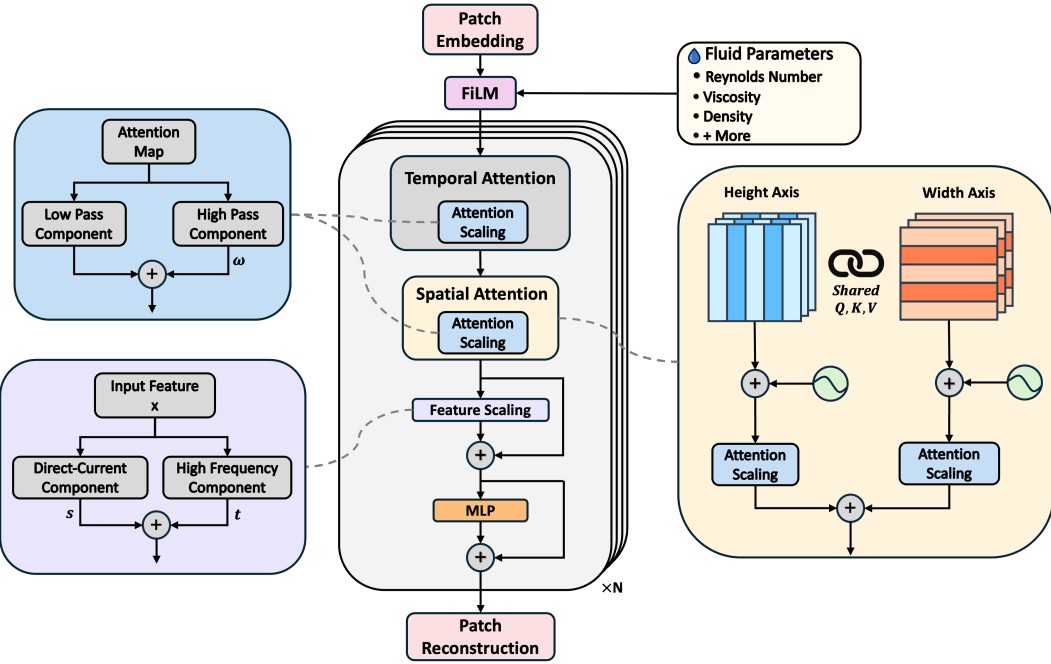

Figure 3: **Bubbleformer architecture.** Input fields are encoded into multiscale spatiotemporal patches, conditioned with fluid-specific parameters, transformed by frequency-scaled temporal and spatial axial attention, and decoded into future velocity, temperature, and interface fields.

**Hierarchical Patch Embedding.** Bubbleformer applies a hierarchical MLP (hMLP) stem inspired by recent hybrid vision transformers [50] to embed input physical fields as spatiotemporal patches. Each time slice is processed using a series of non-overlapping 2×2 convolutional layers with stride 2, progressively reducing spatial resolution and increasing representational depth. This design differs from prior hMLP implementations that fix resolution to 16×16 patches [50, 41]. Instead, we stack repeated 2×2 kernels at each stage to support flexible generalization across multiple resolutions. Compared to flat patch tokenization in ViT [14], this embedding builds a multiscale feature hierarchy with a stronger inductive bias for boiling dynamics, especially in domains with high aspect-ratio and varying discretizations.

**FiLM-based Parameter Conditioning.** To generalize across fluids (such as cryogens, refrigerants, and dielectrics), Bubbleformer conditions embedded feature representations on physical parameters. A feature-wise linear modulation (FiLM) layer [44] applies a learned affine transformation on each channel of the patch embeddings using coefficients derived from a 9D fluid descriptor: Reynolds number, Prandtl number, Stefan number, viscosity, density, thermal conductivity, specific heat capacity, heater temperature and nucleation wait time. A small MLP maps this descriptor to channel-wise scaling and bias terms $(\gamma_c, \beta_c)$, which are applied as: $f'_c = \gamma_c f_c + \beta_c$ for each channel $c$. This

conditioning allows the model to adapt its internal representations to fluid-specific thermophysical properties, essential for generalization across working fluids. Without such conditioning, the model cannot distinguish between fluids with drastically different boiling characteristics (see Figure 1) resulting in mispredicted nucleation timing, incorrect interface velocities, and failure to preserve heat flux scaling during inference.

**Factorized Space-Time Axial Attention.** To model long-range spatiotemporal dynamics in boiling flows, Bubbleformer combines *factorized space-time attention* [3, 1] with *axial attention* [27, 28, 52]. In factorized space-time attention, temporal attention is applied first independently at each spatial location followed by spatial attention. This preserves the spatial locality during temporal modeling and has proven effective in video transformers [3, 1]. At each Bubbleformer block, temporal attention operates across input timesteps to learn dynamics such as bubble growth, departure, and renucleation. The resulting encoding from temporal attention is then passed to spatial attention. Instead of 2D self-attention, axial attention further decomposes the temporally updated features into two 1D attentions along the height and width axes. This decomposition reduces the overall computational complexity from $\mathcal{O}(HWT)^2$ in joint space-time attention to $\mathcal{O}(H^2 + W^2 + T^2)$, while still maintaining a global receptive field. This design, successfully applied to PDE forecasting [41, 60] addresses limitations in prior architectures. Direction-aware attention captures anisotropic structure–important for flow boiling–where strong gradients develop in the flow direction. Unlike FNO, which mixes spatial modes globally, axial attention preserves local structure and mitigates spectral oversmoothing. Each axis learns shared query, key, and value projections, and we apply T5 relative position embeddings [45] along both temporal and spatial attention blocks.

**Attention and Frequency Scaling.** To prevent loss of detail in deeper layers, Bubbleformer incorporates frequency-aware attention and feature scaling [53]. Deep transformers often suffer from attention collapse, where repeated application of softmax attention suppresses high-frequency signals, effectively acting as a low-pass filter. This degradation is particularly detrimental in boiling with high-frequency features such as sharp interfaces and condensation vortices. Attention scaling helps mitigate this by modifying attention to act more like an all-pass filter, by separately scaling the low- and high-frequency components of the attention scores. In addition, we also add a feature scaling layer [53] to each spatiotemporal block that explicitly separates the output feature map into low- and high-frequency components, reweights them with separate learnable parameters before recombining them. This acts as an adaptive sharpening filter that helps preserve fine-grained structures essential for modeling phase interfaces and turbulence. Similar high-frequency feature scaling has been applied successfully to improve temperature field prediction in BubbleML datasets using ResUNet and diffusion models [32].

**Patch Reconstruction.** The patch reconstruction mirrors hierarchical patch embedding in reverse. The spatiotemporal output embeddings are progressively upsampled through $k$ transposed convolution layers to recover the original spatial resolution. This reconstruction produces future predictions of all physical fields–temperature, velocity, and signed distance–at each output timestep.

## 4.2 Metrics

We adopt and extend the metrics introduced in BubbleML [25] to evaluate both short-term predictive accuracy and long-term physical fidelity. BubbleML includes field-based metrics, such as root mean squared error (RMSE), maximum squared error, relative L2 error, boundary RMSE (BRMSE), RMSE along bubble interfaces (IRMSE), and low/mid/high Fourier mode errors. These metrics provide a comprehensive view of spatial and frequency-domain accuracy, particularly useful for evaluating sharp gradients (e.g., temperature discontinuities across liquid-vapor boundaries) that may be masked by global averaging. To evaluate Bubbleformer's long-horizon forecasting, we introduce three additional physically interpretable system-level metrics.

**Heat Flux Consistency.** Boiling is inherently chaotic, and Flash-X simulations represent one possible trajectory of bubble dynamics under given boundary conditions among many. Small deviations in predicted bubble dynamics may lead to increasingly dissimilar fields, yet still preserve physically plausible behavior. To assess system-level consistency, we measure the heat flux distribution across the heater surface over time.

In thermal science, *heat flux* is a system-level indicator of boiling efficiency and its peak value–*critical heat flux* (CHF)–marks the transition to boiling crisis due to the formation of a vapor barrier (see

Figure 1). Accurate prediction of heat flux is essential for safe and efficient design of heat transfer systems [39, 46, 61, 49]. We compute heat flux normal to the heater surface using Fourier's law [30]:

$$q^j = k_\ell \left. \frac{\partial T^j}{\partial n} \right|_{\text{wall}}, \quad j \in \{t, \ldots, t+k\}$$

where $k_l$ is the thermal conductivity of the liquid and $T$ is the temperature field. We accumulate heat flux over time, estimate its empirical distribution ($P_{\text{GT}}(q)$ and $P_{\text{ML}}(q)$) using kernel density estimation, and report mean, standard deviation, and Kullback-Leibler (KL) divergence [4] between simulation and model distributions:

$$D_{KL}(P_{\text{GT}} \| P_{\text{ML}}) = \int P_{\text{GT}}(q) \log \frac{P_{\text{GT}}(q)}{P_{\text{ML}}(q)} \, \mathrm{d}q$$

**Eikonal Loss.** Phase interfaces (i.e., bubble positions) are represented via a signed distance field $\phi(x, y)$, which must satisfy the Eikonal equation: $|\nabla \phi| = 1$ throughout the domain. To assess geometric correctness of the predicted interfaces, we compute the pointwise Eikonal residual [57] and report its average across all $N$ grid points:

$$\mathcal{L}_{eik}(\phi) = \frac{1}{k} \sum_{j=t}^{t+k} \frac{1}{N} \sum_{i=1}^{N} \left| |\nabla \phi_{\text{ML}}^j(x_i)| - 1 \right| \tag{1}$$

A low Eikonal loss indicates that predicted interfaces preserve the level set property of the signed distance field and conform to physically valid bubble geometries. In practice, we find that losses below 0.1 threshold are sufficient to ensure stable interface evolution under autoregressive rollout.

**Mass Conservation.** In boiling systems with fixed heater temperature, fluid properties, and surface geometry, the total vapor volume should remain approximately conserved, up to fluctuations from nucleation and condensation. We assess this, we compute the deviation in vapor volume between model predictions and ground truth simulations across the rollout window $[T, T + k]$. We define the relative vapor volume error as:

$$\mathcal{L}_{vol}(\phi) = \frac{1}{k} \sum_{j=t}^{t+k} \frac{\left| \sum_{i=1}^{N} \mathbf{1}_{\{\phi_{\text{ML}}^j(x_i) > 0\}} - \sum_{i=1}^{N} \mathbf{1}_{\{\phi_{\text{GT}}^j(x_i) > 0\}} \right|}{\sum_{i=1}^{N} \mathbf{1}_{\{\phi_{\text{GT}}^j(x_i) > 0\}}}$$

where $\mathbf{1}_{\{\phi(x) > 0\}}$ is an indicator function for the vapor region, inferred from the signed distance field $\phi(x)$ at each timestep. A low $\mathcal{L}_{vol}(\phi)$ indicates that the model conserves global vapor volume in the domain.

## 5  BubbleML 2.0 Dataset

BubbleML 2.0 expands the original BubbleML dataset [25] with new fluids, boiling configurations, and flow regimes, enabling the study of generalization across thermophysical conditions and geometries. It adds **over 160 new high-resolution 2D simulations** spanning pool boiling and flow boiling, with diverse physics including saturated, subcooled, and single-bubble nucleation across three fluid types: FC-72 (dielectric), R-515B (refrigerant), and $LN_2$ (cryogen). In addition to fluid diversity, BubbleML 2.0 introduces new constant heat flux boundary conditions with double-sided heaters to simulate different boiling regimes, including bubbly, slug, and annular until dryout. Simulating these phenomena required advances in numerical methods in Flash-X. Appendix A provides a detailed description of the simulations and validates against experimental data.

Table 1 summarizes the dataset. All simulations are performed using Flash-X and stored in HDF5 format. Spatial and temporal resolution vary across fluids based on differences in characteristic scales, and adaptive mesh refinement (AMR) is used where needed for computational efficiency. Simulations performed on AMR grids are interpolated to regular grids with the same discretization as the other datasets. We first apply linear interpolation, followed by nearest-neighbor interpolation to resolve boundary NaN values. Additional details, including boundary conditions and data access instructions, are provided in Appendix B.

Table 1: **Summary of BubbleML 2.0 datasets and their parameters.** $\Delta t$ is the temporal resolution in non-dimensional time which depends on the characteristic length and velocity for each fluid, as calculated in Table 3. PB: pool boiling. FB: flow boiling.

| Type - Physics - Fluid | Sims | Domain $(mm^d)$ | Resolution Spatial | $\Delta t$ | Time steps | Size (GB) |
|---|---|---|---|---|---|---|
| PB - Single Bubble - FC72 | 11 | $4.38 \times 6.57$ | $192 \times 288$ | 0.2 | 2000 | 50 |
| PB - Single Bubble - R515B | 11 | $6.48 \times 9.72$ | $192 \times 288$ | 0.2 | 2000 | 50 |
| PB - Saturated - FC72 | 20 | $11.68 \times 11.68$ | $512 \times 512$ | 0.1 | 2000 | 320 |
| PB - Saturated - R515B | 20 | $17.28 \times 17.28$ | $512 \times 512$ | 0.1 | 2000 | 320 |
| PB - Saturated - LN2 | 20 | $16.96 \times 16.96$ | $512 \times 512$ | 0.1 | 2000 | 320 |
| PB - Subcooled - FC72 | 20 | $11.68 \times 11.68$ | $512 \times 512$ | 0.1 | 2000 | 320 |
| PB - Subcooled - R515B | 20 | $17.28 \times 17.28$ | $512 \times 512$ | 0.1 | 2000 | 320 |
| PB - Subcooled - LN2 | 20 | $16.96 \times 16.96$ | $512 \times 512$ | 0.1 | 2000 | 320 |
| FB - Inlet Velocity - FC72 | 15 | $30.66 \times 3.65$ | $1344 \times 160$ | 0.1 | 2000 | 320 |
| FB - Constant Heat Flux - FC72 | 6 | $117.5 \times 5.1$ | $5152 \times 224$ | 0.1 | 1000 | 420 |

BubbleML 2.0 follows FAIR data principles [54] and includes metadata and physical parameters for every simulation as outlined in Appendix B.2. The dataset enables benchmarking of generalization across fluids, boundary conditions, flow regimes, and supports training long-horizon forecasting models such as Bubbleformer.

# 6 Results and Discussion

## 6.1 Training

We train Bubbleformer to forecast future physical fields in boiling systems, including bubble positions (signed distance field $\phi$), temperature field ($T$), and velocity vector field ($\vec{u}$). Given $k$ past frames $[\phi, T, \vec{u}]_{t-k:t-1}$, the model predicts the next $k$ frames $[\phi, T, \vec{u}]_{t:t+k-1}$ in a bundled fashion [5]. Models are trained in a supervised manner by minimizing the sum of relative L2 norms across the predicted physical fields, $[\phi^*, \vec{u}^*, T^*]_{t:t+k-1}$. The single-step loss is given by:

$$\mathcal{L} = \frac{1}{k} \sum_{i=t}^{t+k-1} \left( \frac{||\phi_i - \phi_i^*||_2}{||\phi_i||_2} + \frac{||T_i - T_i^*||_2}{||\phi_i||_2} + \frac{||\vec{u}_i - \vec{u}_i^*||_2}{||\vec{u}_i||_2} \right)$$

We train two Bubbleformer models, *small* (29M parameters) and *large* (115M parameters). using a prediction window of $k = 5$ steps and train for 250 epochs using the Lion optimizer [7] and a warm-up cosine scheduler. Additional architectural and training details, including hyperparameter settings, are provided in Appendix C.

## 6.2 Re-nucleation and Forecasting

In autoregressive forecasting, a basic requirement for all models is the nucleation of new bubbles on the heater surface when old ones depart. While prior models fail to renucleate, Bubbleformer successfully learns this behavior, capturing the temporal dynamics required for periodic bubble formation and maintaining stable rollouts over extended horizons.

We observe that the re-nucleation process learned by Bubbleformer is stochastic: predictions initially align closely with simulations, but diverge gradually over time due to the chaotic nature of the system. Despite this divergence, the model's forecasts continue to respect physical laws. As shown in Figure 4, over a 200-step rollout on an unseen flow boiling trajectory, Bubbleformer predicted fields remain physically well-behaved. The system level quantities are conserved: the predicted heat flux distribution remain close to that of the simulation, the bubble positions are valid signed distance fields, and mass conservation is closely followed. These results show that Bubbleformer does not merely replicate a simulation trajectory, but learns to generate valid realizations of the underlying boiling process.

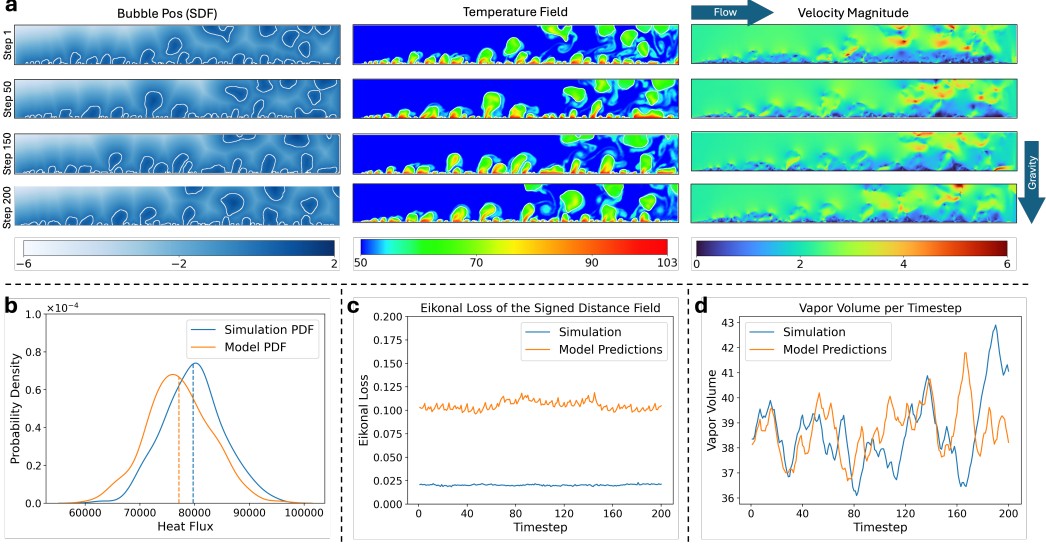

Figure 4: **Flow Boiling Forecasting.** (a) Rollout for a Bubbleformer-S model on an unseen flow boiling trajectory (FC-72, input flow scale = 2.2). (b) Comparison of predicted vs. ground-truth heat flux PDFs. (c) Per-frame Eikonal loss of forecasted signed distance fields representing bubble positions. (d) Vapor volume in the boiling domain over time for both model and simulation.

Additional discussion of all forecasting models are provided in Appendix C. We also validate our proposed metrics on a deterministic single-bubble simulation study in Appendix C.3.

### 6.3 Prediction of Velocity and Temperature

We evaluate Bubbleformer on the supervised prediction task introduced in BubbleML [25], modeling temperature and velocity fields in subcooled pool boiling. As shown in Figure 5, Bubbleformer outperforms the best-performing models in BubbleML– UNet$_{mod}$ and Factorized Fourier Neural Operator (FFNO)–achieving state-of-the-art accuracy across all reported metrics, including relative L2 error. To assess long-horizon prediction stability, we extend the autoregressive rollout from 400 to 800 timesteps–doubling the original evaluation setting in BubbleML. Bubbleformer maintains accurate predictions over this extended horizon, while baseline models exhibit growing instability.

We attribute Bubbleformer's performance to its spatiotemporal attention, which enables it to resolve both sharp, non-smooth interfaces characteristic of boiling flows and long-range dependencies. While FFNO and UNet$_{mod}$ suffer large error spikes during violent bubble detachment events and exhibit a steadily growing error thereafter, Bubbleformer maintains a uniformly low error across the entire 800-step rollout. Notably, it remains stable even up to 2000 rollout steps, this domenstrates that our spatialtemporal attention architecture is robust to localized topological changes and yields stable long-horizon predictions. A complete listing of error metrics for each model and dataset pairing can be found in Appendix D.

## 7 Conclusions and Limitations

We introduce Bubbleformer, a spatiotemporal transformer for forecasting boiling dynamics across fluids, geometries, and regimes. Bubbleformer integrates axial attention, frequency-aware scaling, and fluid-conditioned FiLM layers to jointly predict and forecast velocity, temperature, and interface fields. Our results show that Bubbleformer outperforms prior models in both short-term accuracy and long-horizon stability, while preserving physical consistency as measured by system-level metrics such as heat flux, Eikonal loss, and vapor mass conservation. To support generalization, we introduce BubbleML 2.0–a diverse, high-resolution dataset from over 160 simulations.

**Limitations.** The current Bubbleformer architecture cannot natively operate on AMR grids, necessary for simulating fluids such as water or large real-world domains. Interpolation to uniform grids can

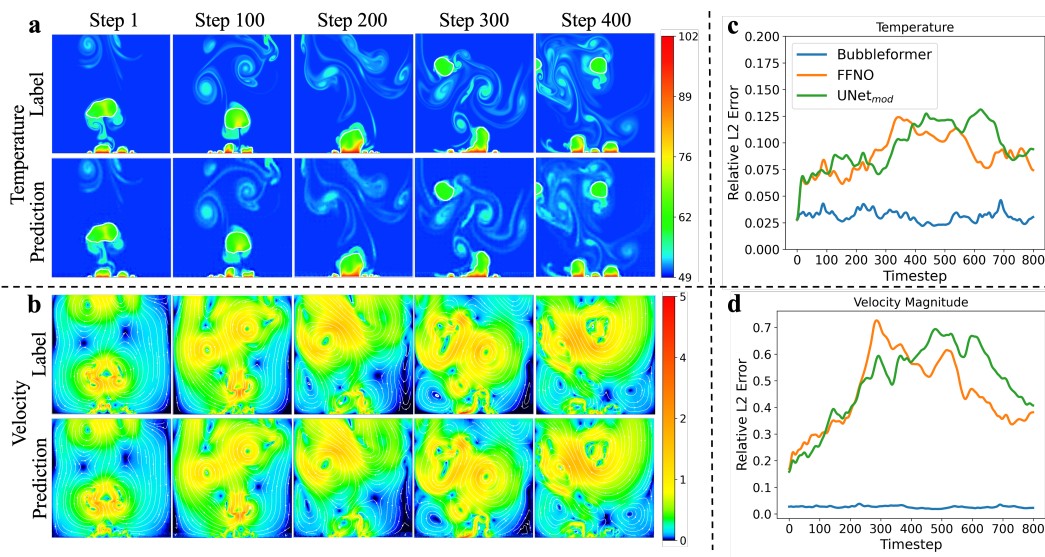

Figure 5: **Subcooled Pool Boiling Prediction.** (a) and (b) Predicted temperature and velocity fields from the Bubbleformer-S model on an unseen subcooled pool boiling trajectory for FC-72. (c) and (d) Relative L2 error over 800 rollout timesteps for temperature and velocity magnitude (combined x and y components), comparing Bubbleformer-S, FFNO, and UNet$_{mod}$.

introduce numerical errors that can make model training unstable. Extending the model to directly support AMR inputs–via hierarchical encoding or point-based attention–is an important direction.

Current models are specialized for one type of physics. Combining datasets from different physics (e.g., subcooled and saturated pool boiling) remains challenging. Incorporating patch-level routing via mixture-of-experts models may improve scalability and generalization.

Training on high-resolution datasets requires a large amount of GPU memory, primarily due to the storage of activations during backpropagation. While activation checkpointing can alleviate memory usage, it increases computation time. Neural operators, with their resolution-invariant properties, offer a potential solution by enabling training at lower resolutions. However, their current formulations struggle to capture the sharp discontinuities associated with nucleation events. Advancing neural operators to handle such features is a promising direction for future research in forecasting boiling dynamics.

## Acknowledgments and Disclosure of Funding

The authors gratefully acknowledge funding support from the Office of Naval Research(ONR) MURI under Grant No. N000142412575, with Dr. Mark Spector serving as the program officer. We also thank the Research Cyberinfrastructure Center at the University of California, Irvine, for the GPU computing resources on the HPC3 cluster. This work was partially supported by the Scientific Discovery through Advanced Computing (SciDAC) program via the Office of Nuclear Physics and the Office of Advanced Scientific Computing Research in the Office of Science at the U.S. Department of Energy, and the Laboratory Directed Research and Development (LDRD) program at Argonne National Lab under contract number DE-AC02-06CH11357.

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
