# A  Numerical Formulation

Boiling occurs when a liquid undergoes evaporation on the surface of a solid heater, resulting in formation of gas (vapor) bubbles which induce turbulence and improve heat transfer efficiency. The dynamics of the bubbles is governed by the balance f forces: gravity (buoyancy) pulling the dense liquid downward and pushing vapor upward, surface tension ($\sigma$) trying to minimize the liquid–vapor interface area, and evaporative heat flux. In our numerical model, the liquid-gas interface $\Gamma$ is tracked using a level-set $\phi$, a signed distance function that is positive inside the gas and negative inside the liquid. $\phi = 0$, represents the implicit location of $\Gamma$.

We solve a coupled system of incompressible Navier–Stokes equations and energy equations in each phase. Assuming an incompressible flow, the governing equations can be described for each phase using tensor notations for a 3D domain.

The equations for *liquid phase* (subscript $L$) are written as:

$$\frac{\partial \vec{\mathbf{u}}}{\partial t} + (\vec{\mathbf{u}} \cdot \nabla)\vec{\mathbf{u}} = -\frac{1}{\rho'_L}\,\nabla P \;+\; \frac{1}{\text{Re}}\,\nabla \cdot \left(\frac{\mu'_L}{\rho'_L}\,\nabla\vec{\mathbf{u}}\right) \;+\; \frac{\vec{\mathbf{g}}}{\text{Fr}^2} \tag{2a}$$

$$\frac{\partial T}{\partial t} + \vec{\mathbf{u}} \cdot \nabla T = \frac{1}{\rho'_L\,C'_{pL}\,\text{Pe}}\,\nabla \cdot \left(k'_L\,\nabla T\right) \tag{2b}$$

For the *gas phase* (subscript $G$), the form is analogous:

$$\frac{\partial \vec{\mathbf{u}}}{\partial t} + (\vec{\mathbf{u}} \cdot \nabla)\vec{\mathbf{u}} = -\frac{1}{\rho'_G}\,\nabla P \;+\; \frac{1}{\text{Re}}\,\nabla \cdot \left(\frac{\mu'_G}{\rho'_G}\,\nabla\vec{\mathbf{u}}\right) \;+\; \frac{\vec{\mathbf{g}}}{\text{Fr}^2} \tag{3a}$$

$$\frac{\partial T}{\partial t} + \vec{\mathbf{u}} \cdot \nabla T = \frac{1}{\rho'_G\,C'_{pG}\,\text{Pe}}\,\nabla \cdot \left(k'_G\,\nabla T\right) \tag{3b}$$

where $\vec{\mathbf{u}}$ is the velocity field, $P$ is the pressure, and $T$ is the temperature everywhere in the domain. The equations are non-dimensionalized with reference quantities from the liquid phase resulting in Reynolds number (Re) = $\rho_L u_0 l_0/\mu_L$, Prandtl number (Pr) = $\mu_L C_{p_L}/k_L$, Froude number (Fr) = $u_0/\sqrt{gl_0}$, and Peclet number (Pe) = Re Pr. By construction, all non-dimensional liquid reference properties are normalized to 1 and each gas property is expressed relative to its liquid counterpart:

$$y'_L = 1, \quad y'_G = \frac{y_G}{y_L} \tag{4}$$

for any property $y \in \rho, \mu, C_p, k$ (density, viscosity, specific heat, and thermal conductivity respectively). $u_0$ and $l_0$ are reference velocity and length scales, and $g$ is acceleration due to gravity. For boiling problems, the reference length scale is set to the capillary length $l_0 = \sqrt{\sigma/(\rho_L - \rho_G)\,g}$ and the velocity scale is the terminal velocity $u_0 = \sqrt{g\,l_0}$. For more details on how these parameters are calculated, we refer the reader to Appendix B. The reference temperature scale is given by $(T - T_{bulk})/\Delta T$, where $\Delta T = T_{wall} - T_{bulk}$ (constant wall temperature) or where $\Delta T = T_{ref} - T_{bulk}$ (constant wall heat flux).

Within each phase, the flow is incompressible (constant density), so any volume change is solely due to phase change at the interface. The continuity equation is written as:

$$\nabla \cdot \vec{\mathbf{u}} = -\,\dot{m}\,\vec{n}_\Gamma \cdot \nabla\frac{1}{\rho'}\,\bigg|_\Gamma \tag{5}$$

where $\vec{n} = \nabla\phi/\sqrt{|\nabla\phi|}$ is a unit vector normal to the liquid-gas interface, and $\dot{m}$ is the evaporative mass flux. Equation 5 states that $\nabla \cdot \vec{\mathbf{u}} = 0$ everywhere except at the interface, where a source term on the right-hand side accounts for the conversion of mass from liquid to vapor or vice versa.

The interfacial mass flux $\dot{m}$ is determined by the jump in heat flux at the interface (i.e. how much thermal energy is being used to produce phase change). We denote by $\vec{n} \cdot \nabla T_L$ the difference between heat flux from liquid to gas, and $\vec{n} \cdot \nabla T_G$ the heat flux gradient from gas to liquid, normal to the liquid-gas interface $\Gamma$. The difference between these heat fluxes determines the mass transfer from one phase to the other. The value of $\dot{m}$ at the interface is calculated using the energy balance:

$$\dot{m} = \frac{\text{St}}{\text{Pe}}\left(\vec{n}_\Gamma \cdot k'_L\,\nabla T_L\,\big|_\Gamma - \vec{n}_\Gamma \cdot k'_G\,\nabla T_G\,\big|_\Gamma\right) \tag{6}$$

where St is the Stefan number defined by $\text{St} = C_{p_L}\Delta T/Q_l$, with $Q_l$ as latent heat of evaporation.

The level-set function representing the interface is computed using the convection equation:

$$\frac{\partial \phi}{\partial t} + \vec{\mathbf{u}}_\Gamma \cdot \nabla\phi = 0 \tag{7}$$

where $\vec{\mathbf{u}}_\Gamma = \vec{\mathbf{u}} + (\dot{m}/\rho')\vec{n}_\Gamma$ is the *interface velocity*. The convection of level-set is accompanied by a selective reinitialization technique [11] to mitigate diffusive errors that are generated from numerical discretization of the convection term. This reinitialization step is essential to preserve accuracy of the interface over long simulations, since the advection equation can cause $\phi$ to lose its signed distance property.

The presence of the interface leads to discontinuities (jumps) in certain field variables. These jump conditions are modeled using a Ghost Fluid Method (GFM) [22, 10].

**Velocity Jump.** Reorganizing equation for $\vec{\mathbf{u}}_\Gamma = \vec{\mathbf{u}} + (\dot{m}/\rho')\vec{n}_\Gamma$ results in the following expression for jump in velocity normal to $\Gamma$:

$$[\vec{\mathbf{u}}]_\Gamma \ = \ \vec{\mathbf{u}}_G - \vec{\mathbf{u}}_L \ = \ \dot{m}\,\vec{n}_\Gamma\Big(\frac{1}{\rho'_G} - \frac{1}{\rho'_L}\Big) \tag{8}$$

**Pressure Jump.** The mass flux $\dot{m}$, surface tension $\sigma$, and viscous stresses contribute towards a similar jump in pressure, where $\kappa$ is the interface curvature. The non-dimensional form for pressure jump using $\text{We} = \rho_l u_0^2 l_0/\sigma$ can be written as:

$$[P]_\Gamma \ = \ P_G - P_L \ = \ \frac{\kappa}{\text{We}} - \Big(\frac{1}{\rho'_G} - \frac{1}{\rho'_L}\Big)\dot{m}^2 \tag{9}$$

The effect of viscous jump is assumed to be negligible in this formulation due to the smeared treatment of viscosity near the interface described in [10]. This assumption is consistent with the formulation in [31].

**Temperature Continuity.** Finally, the boundary condition for temperature at the liquid-gas interface is given by:

$$T_\Gamma = T_{sat} \tag{10}$$

where $T_{sat}$ corresponds to saturation temperature.

We consider two types of thermal boundary conditions in our simulations: *constant wall temperature* and *constant wall heat flux*. For a constant wall temperature case, the heater is assigned a non-dimensional temperature using a Dirichlet boundary condition:

$$T'_{wall} = \frac{T_{wall} - T_{bulk}}{\Delta T} = 1 \tag{11}$$

In contrast, for a constant wall heat flux boundary, the wall temperature is determined using a Neumann boundary condition derived from the specified non-dimensional Nusselt number:

$$\text{Nu}_{wall} = \frac{q\,l_0}{\Delta T\,k_L} \tag{12}$$

where $q$ is the imposed wall heat flux. These boundary conditions provide the thermal driving force at the solid surface and govern the rate of phase change at the liquid-gas interface.

## B    Additional BubbleML 2.0 Details

### B.1    Dataset URLs and Links

**Code:** Code for training and evaluation of all the benchmark models are available at our Bubbleformer GitHub repository. The code is released under open MIT license.

**Dataset:** Our dataset is hosted in Hugging Face at BubbleML 2.0 under the Creative Commons Attribution 4.0 International License. The dataset can be used with either our published Github code or the Hugging Face datasets[37] python package.

**Model Weights:** Weights for all the benchmark models are available in the model zoo directory of the same github repository. All relevant benchmark results can be accessed on the same page.

**DOI:** The BubbleML 2.0 dataset has a DOI from Huggingface: 10.57967/hf/5594 and can be cited using the following citation:

```
@misc{hpcforge_lab_@_uc_irvine_2025,
 author      = { HPCForge Lab @ UC Irvine and Sheikh Md Shakeel Hassan and
                          Xianwei Zou and Akash Dhruv and Vishwanath Ganesan
                          and Aparna Chandramowlishwaran },
 title       = { BubbleML_2 (Revision 2307458) },
 year        = 2025,
 url         = { https://huggingface.co/datasets/hpcforge/BubbleML_2 },
 doi         = { 10.57967/hf/5594 },
 publisher   = { Hugging Face }
}
```

**Documentation and Tutorials:** We provide detailed documentation (as README files) and Jupyter notebook examples in our GitHub repository to load our data, train a model, perform inference using a trained model, and visualize results.

## B.2 Maintenance and Long Term Preservation

The authors are committed to maintaining and preserving this dataset. We have closely worked with Hugging Face to ensure the long term availability of the dataset with the added possibility of extending the dataset without running into storage limits.

**Findable:** All data is stored in Hugging Face. All present and future data will share a global and persistent DOI 10.57967/hf/5594.

**Accessible:** All data and descriptive metadata can be downloaded from Hugging Face using their API or Git.

**Interoperable:** The data is provided in the form of standard HDF5 files that can be read using many common libraries, such as h5py for Python. Our codebase contains well documented code to load the dataset for various downstream tasks. Relevant metadata is stored as json files with the same name as the simulation. We also provide flexibility to the user to load our data using the Hugging Face datasets[37] library.

**Reusable:** BubbleML 2.0 is released under the Creative Commons Attribution 4.0 International License. Bubbleformer codebase is released under the MIT License.

## B.3 Contact Line Dynamics

In simulations involving multiple interacting vapor bubbles, realistic modeling of the contact line is essential to accurately capture the bubble dynamics and phase change phenomena. Experimental observations indicate that the contact angle at the bubble base transitions between two limiting values—the receding and advancing contact angles—depending on the direction of contact line motion.

One prevailing interpretation attributes this behavior to inertial effects induced by phase change near the heated surface. During spontaneous evaporation, momentum generated in the direction of vapor departure causes the contact line to recede. When buoyancy forces begin to influence the bubble, the direction of contact line motion reverses. Inertia, now opposing this motion, leads to a gradual increase in the contact angle until the advancing motion dominates and the bubble base begins to shrink. These inertial effects are primarily governed by wall adhesion forces and significantly impact phenomena such as bubble coalescence, where sudden momentum transfers disturb the contact line equilibrium.

To model these effects, Continuum Surface Force (CSF) approaches incorporate the direct contribution of wall adhesion to the liquid-vapor interface force balance [58, 47, 48]. In the case of sharp-interface methods such as level set techniques, the contact line is enforced by prescribing a contact angle

boundary condition for the level set function. Inertial effects are thus embedded in this boundary condition by dynamically adjusting the local contact angle based on the near-wall flow.

In our implementation, the instantaneous contact angle $\psi$ is defined as a piecewise function of the near-wall radial velocity $u_{base}$ in the plane of the heater surface:

$$\psi = \begin{cases} \psi_r, & \text{if } u_{base} < 0 \\ \frac{\psi_a - \psi_r}{u_{lim}} u_{base} + \psi_r, & \text{if } 0 \leq u_{base} \leq u_{lim} \\ \psi_a, & \text{if } u_{base} > u_{lim} \end{cases} \tag{13}$$

Here, $\psi_r$ and $\psi_a$ represent the receding and advancing contact angles, respectively; $u_{lim}$ is the limiting velocity beyond which the contact angle saturates. This formulation is adapted from the model proposed by Mukherjee et al. [43], with the key distinction that our approach attributes inertial effects solely to the advancing motion of the contact line. We find that an optimal value for $u_{lim}$ lies between 20% and 25% of the characteristic velocity, defined as $u_c$. A receding angle of $\psi_r = \pi/4$ is chosen to maintain a balance between surface tension, inertial, and gravitational forces, given that $\sin(\pi/4) = \cos(\pi/4)$. The advancing angle $\psi_a$ is treated as a function of local wall temperature or heat flux.

## B.4 Nucleation Site Distribution and Dynamics

The distribution and density of nucleation sites on the heater surface are critical parameters for replicating experimental boiling regimes. As wall superheat increases, the number of active nucleation sites typically rises, leading to transitions between boiling regimes on the heat transfer curve [29].

Several models exist for estimating nucleation site density, ranging from empirical correlations based on observed data [23] to formulations based on surface roughness characteristics [58]. In our framework, we initialize the nucleation site distribution using experimental estimates of bubble density (in bubbles/mm$^2$). These sites are then spatially assigned using a quasi-random low-discrepancy Halton sequence to avoid artificial clustering while maintaining reproducibility.

As wall superheat or heat flux increases during simulation, additional nucleation sites are introduced while preserving the initial configuration. This approach allows us to match experimentally observed bubble site densities without relying heavily on empirical models.

Once the nucleation map is initialized, it acts as a template for bubble generation. At each timestep, the model checks whether the four cells surrounding a nucleation site are filled with liquid. If so, the site is marked for re-nucleation after a specified waiting time $t^*_{wait}$. If vapor reoccupies any of these cells before the wait period elapses, the re-nucleation flag is reset. Newly nucleated bubbles are assigned an initial radius of $0.1\, l_c$.

This dynamic nucleation model allows for physically realistic bubble generation patterns and facilitates the study of transient effects such as intermittent boiling and coalescence-driven heat transfer enhancement.

## B.5 Bubble Wait Time Modeling

$t^*_{\text{wait}}$ is a critical parameter in the nucleation site model, representing the delay between successive bubble departures at a given site. Its value governs the frequency of nucleation events and thus directly affects heat transfer rates and bubble interactions in the simulation. The modeling of wait time depends on the thermal boundary condition applied to the wall: constant heat flux or constant wall temperature.

### B.5.1 Wait Time for Constant Heat Flux

For constant heat flux simulations, the bubble frequency is modeled following the heat flux partitioning approach proposed by Basu et al. [2]. The frequency $f$ of bubble departure from a nucleation site is calculated using the relationship:

$$f = \frac{q}{N_d E} \tag{14}$$

where, $q$ is the applied wall heat flux, $N_d$ is the nucleation site density (bubbles/mm$^2$), $E$ is the energy removed per bubble, given by:

$$E = \frac{4\pi}{3} \rho_v h_{lv} R_d^3 \tag{15}$$

with $\rho_v$ being the vapor density, $h_{lv}$ the latent heat of vaporization, and $R_d$ the bubble departure radius.

After non-dimensionalization using the capillary length $l_c$ and characteristic velocity $u_c = \sqrt{g_e l_c}$, the inverse bubble frequency becomes:

$$\frac{1}{f} = R_d^3 \cdot \left( \frac{4\pi N_d}{3} \right) \cdot \frac{\rho^* \operatorname{Re} \operatorname{Pr}}{\operatorname{St} \cdot \operatorname{Nu}_{\text{wall}}} \tag{16}$$

Here, $R_d$ is non-dimensional and computed using an empirical relation adapted from Han and Griffith [8]:

$$R_d = 0.4251 \, \psi \, \sqrt{2 \operatorname{Bo}} \tag{17}$$

where $\psi$ is the static contact angle in radians and $\operatorname{Bo}$ is the Bond number based on $l_c$.

The total bubble life cycle is partitioned into growth and waiting phases, based on empirical assumptions from Xiao et al. [56]:

$$t_{\text{growth}}^* = \frac{1}{4f}, \qquad t_{\text{wait}}^* = \frac{3}{4f} \tag{18}$$

This approach ensures that the nucleation frequency aligns with both energy removal and bubble dynamics under steady heat flux conditions. A Python script to perform this apriori calculation is available in our lab-notebooks [13].

### B.5.2 Wait Time for Constant Wall Temperature

For simulations with a prescribed wall temperature (Dirichlet condition), the bubble dynamics are not directly driven by an imposed heat flux. Instead, we estimate the characteristic time for re-nucleation based on the time a bubble requires to traverse a distance equal to the capillary length $l_c$ at terminal velocity $u_c$. This time scale,

$$t_c = \frac{l_c}{u_c}, \tag{19}$$

represents the dominant dynamical time for bubble departure under gravity and surface tension balance. Accordingly, the nucleation wait time for constant wall temperature cases is selected to be:

$$t_{\text{wait}}^* \in [0.4, \ 0.6, \ 1.0] \, t_c \tag{20}$$

for FC-72, R515B and LN2 respectively. This range accounts for slight variations in bubble terminal dynamics due to interactions or residual vapor near the wall, while maintaining consistency with the overall phase change timescale.

### B.6 Fluids

We can broadly classify the fluids into cryogens, refrigerants, fluorochemicals, and water based on saturation temperature. Depending on the boiling point, a cryogen is utilized for space applications and cooling superconducting motors, whereas low-temperature refrigerants are used in evaporators and heat exchangers in HVAC&R industries. Similarly, to maintain the operational temperature of ultra-high heat flux chips at nominal temperature ranges in HPC datacenters, room temperature fluids such as dielectrics and water are used for direct immersion-based and indirect cold plate-based thermal management systems for electronic cooling. Due to fluid physics unique to these distinct classes of working fluids, these fluids also exhibit distinct thermophysical properties, leading to distinct boiling heat transfer and two-phase flow behavior [17, 18, 20].

BubbleML 1.0 dataset includes one dielectric fluid. However, to account for distinct fluid physics unique to cryogens, refrigerants, fluorochemicals, and water, in 2.0, we expand the dataset to other working fluids and include at least one from each class of working fluid to cover a wide range of thermodynamic and fluid properties summarized in Table 2.

| Parameters | Units | Water | FC-72 | R515b | LN$_2$ |
|---|---|---|---|---|---|
| Saturation Temperature ($T_{sat}$) | °C | 100 | 58 | -19 | -196 |
| Liquid Density ($\rho_l$) | kg·m$^{-3}$ | 958.35 | 1575.6 | 1313.7 | 807 |
| Vapor Density ($\rho_v$) | kg·m$^{-3}$ | 0.5982 | 13.687 | 5.8361 | 4.51 |
| Liquid Viscosity ($\mu_l$) | N·s·m$^{-2}$ | $2.82\times10^{-4}$ | $4.18\times10^{-4}$ | $3.427\times10^{-4}$ | $1.62\times10^{-4}$ |
| Vapor Viscosity ($\mu_v$) | N·s·m$^{-2}$ | $1.232\times10^{-5}$ | $1.177\times10^{-5}$ | $9.626\times10^{-6}$ | $5.428\times10^{-6}$ |
| Liquid Specific Heat Capacity ($C_{p_l}$) | J·kg$^{-1}$·K$^{-1}$ | 4215.7 | 1099.5 | 1263.6 | 2040.5 |
| Vapor Specific Heat Capacity ($C_{p_v}$) | J·kg$^{-1}$·K$^{-1}$ | 2080 | 879.30 | 823.26 | 1122.4 |
| Liquid Thermal Conductivity ($k_l$) | W·m$^{-1}$·K$^{-1}$ | 0.677 | $6.25\times10^{-2}$ | $8.887\times10^{-2}$ | 0.145 |
| Vapor Thermal Conductivity ($k_v$) | W·m$^{-1}$·K$^{-1}$ | $2.457\times10^{-2}$ | $1.306\times10^{-2}$ | $1.029\times10^{-2}$ | $7.163\times10^{-3}$ |
| Latent Heat of Vaporization ($h_{lv}$) | J·kg$^{-1}$ | $2.256\times10^{6}$ | $8.4227\times10^{4}$ | $1.9056\times10^{5}$ | $1.9944\times10^{5}$ |
| Surface Tension ($\sigma$) | N·m$^{-1}$ | $5.891\times10^{-2}$ | $8.112\times10^{-3}$ | $1.499\times10^{-2}$ | $8.926\times10^{-3}$ |

Table 2: Thermophysical properties of different fluids at saturation under 1 atm.
Sources: NIST Reference Fluid Thermodynamic and Transport Properties Database [36]

The non-dimensional parameters for the simulations of different fluids are in Table 3. The different thermophysical properties (saturation temperature, liquid/vapor density ratio, surface tension, latent heat of vaporization) impact the physics governing the flows under study in interesting ways. For instance, the low saturation temperature of refrigerants and cryogens affects bubble movement in a way different then that for dielectrics, due to flow of cold liquid towards the heater surface, we can see the bubbles slide on the heater surface befor departure. For instance, owing to low surface tension and latent heat of vaporization, cryogens have been shown to exhibit nucleate boiling dominance at relatively lower wall superheats [17], leading to a large number of smaller-sized bubbles nucleating as opposed to a relatively small number of larger-sized bubbles nucleating for other room temperature fluids. These unique phenomena cause interesting challenges in the learning of data-driven models.

| Simulation Parameter | Formula | Water | FC-72 | R515b | LN$_2$ |
|---|---|---|---|---|---|
| Characteristic Length ($l_c$) (mm) | $\sqrt{\frac{\sigma}{(\rho_l-\rho_v)g}}$ | 2.5 | 0.73 | 1.08 | 1.06 |
| Characteristic Velocity ($u_c$) (m·s$^{-1}$) | $\sqrt{gl_c}$ | 0.16 | 0.08 | 0.1 | 0.1 |
| Characteristic Time ($t_c$) (ms) | $\frac{l_c}{u_c}$ | 16 | 8.6 | 10.5 | 10.4 |
| Density ($\rho'$) | $\frac{\rho_v}{\rho_l}$ | $6.242\times10^{-4}$ | $8.687\times10^{-3}$ | $4.442\times10^{-3}$ | $5.589\times10^{-3}$ |
| Viscosity ($\mu'$) | $\frac{\mu_v}{\mu_l}$ | $4.369\times10^{-2}$ | $2.816\times10^{-2}$ | $2.809\times10^{-2}$ | $3.351\times10^{-2}$ |
| Thermal Conductivity ($k'$) | $\frac{k_v}{k_l}$ | $3.629\times10^{-2}$ | $2.09\times10^{-1}$ | $1.158\times10^{-1}$ | $4.94\times10^{-2}$ |
| Specific Heat ($C_p'$) | $\frac{C_{pv}}{C_{pl}}$ | $4.934\times10^{-1}$ | $7.997\times10^{-1}$ | $6.515\times10^{-1}$ | $5.501\times10^{-1}$ |
| Reynolds Number (Re) | $\frac{\rho_l u_c l_c}{\mu_l}$ | 1334 | 231.72 | 426.67 | 542.13 |
| Weber Number (We) | $\frac{\rho_l u_c^2 l_c}{\sigma}$ | 1.0 | 1.0 | 1.0 | 1.0 |
| Prandtl Number (Pr) | $\frac{\mu_l C_{pl}}{k_l}$ | 1.756 | 7.35 | 4.87 | 2.28 |
| Stefan Number (St) | $\frac{C_{pl}}{h_{lv}}\Delta T$ | $0.0019\Delta T$ | $0.013\Delta T$ | $0.0066\Delta T$ | $0.0102\Delta T$ |

Table 3: Non-dimensional simulation parameters for different fluids at saturation under 1 atm. $\Delta T$ is the difference between the maximum and minimum temperatures of the system (heater temperature and bulk liquid temperature)

For simulation of subcooled and saturated pool boiling of different fluids, we keep ($T_{heater} - T_{bulk}$) consistent across fluids. The bubble growth rate usually scales with Jakob number (in our case St number) that scales with $T_w - T_{sat}$ [59]. Hence, at least for pool boiling, by fixing $T_w - T_{sat}$, we can do a parametric analysis on the bubble growth, merging, and departure dynamics across fluids (effect of thermophysical properties) for the same operating pressure (1 atm) but resulting in different wall heat fluxes (controlled variable in an experiment) due to different HTCs. We choose $T_w$ for

the fluids to be greater than $T_{w,ONB}$ (wall temperature required for boiling incipience). Also the corresponding wall heat flux ($q$) should obey $q_{ONB} < q < q_{CHF}$, where the former serves as the lower limit for boiling incipience and the latter determines the upper limit for critical heat flux.

## B.7    Dataset Validation

**Flow Boiling with Constant Heat Flux.** To validate flow boiling with a constant heat flux boundary

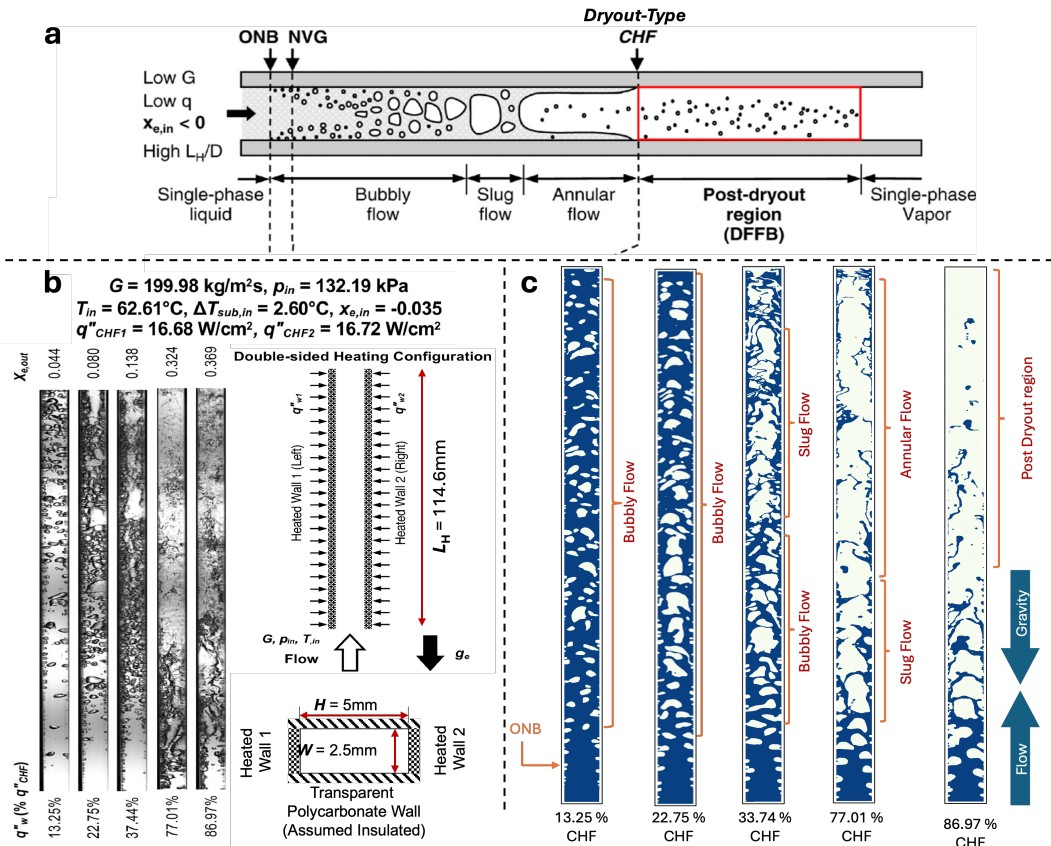

Figure 6: **Comparison of flow boiling simulations at constant heat flux with experimental observations.** (a) Illustration of a horizontal channel showing different flow boiling regimes [19]. (b) Experimental visualizations of FC-72 at increasing heat flux values expressed as a fraction of the critical heat flux (CHF) [9], alongside corresponding 2D simulation domain and boundary conditions. (c) Simulation outputs using Flash-X replicate key features of the flow regimes observed experimentally, including bubbly, slug, annular, and post-dryout flow patterns. Agreement between simulation and experiment validates the fidelity of the numerical framework for modeling flow boiling dynamics.

condition, we employed Flash-X to reproduce a parametric study [9] of steady-state flow boiling under vertical upflow orientation in a rectangular channel with double-sided heating configuration. The experimental setup used FC-72 as the working fluid, with a saturation temperature of 62°C and a bulk inlet temperature of 60°C, resulting in a moderately subcooled or near-saturated inlet.

Simulations were conducted in a two-dimensional rectangular channel domain measuring 118mm × 5mm, discretized using a block-structured AMR mesh with up to three levels of refinement. This grid configuration ensures adequate resolution of thermal boundary layers and vapor-liquid interfaces while maintaining computational efficiency. Boundary conditions mirrored those in the physical experiment: a velocity-driven inflow on the bottom (x-direction), an outflow on the top, and no-slip walls on all other boundaries. A constant heat flux boundary condition was applied to the walls, set as a percentage of the critical heat flux (CHF).

The fluid properties were defined based on FC-72 specifications. Both liquid and vapor phases were modeled using a multiphase formulation with carefully scaled density and viscosity ratios, as well as other thermophysical parameters, detailed in [13].

Gravity was applied in the negative x-direction to reflect the vertical upflow configuration. The simulations used second-order time integration and high-resolution advection via a fifth-order WENO scheme, along with incompressible Navier-Stokes dynamics. A face-centered, divergence-free AMReX interpolator was used to ensure mass conservation across refinement levels. Adaptive mesh refinement (AMR) was triggered using a level-set function to track interface evolution.

Special consideration was given to seeding effects, bubble nucleation timing, and the number of active nucleation sites, which were evaluated a priori and provided as input parameters. While detailed quantitative comparisons of nucleation frequency and bubble dynamics are reserved for future work, the Flash-X simulations show strong qualitative agreement with experimental observations, as illustrated in Figure 6.

## C  Forecasting

### C.1  Model Configurations

We train two Bubbleformer models, details of which are given in Table 4. Bubbleformer-S is a smaller variant with reduced embedding and MLP dimensions, while Bubbleformer-L is a larger model with higher capacity. Both models use FiLM conditioning with 9-dimensional thermophysical inputs and axial attention with 12 transformer blocks and $16{\times}16$ spatiotemporal patch embeddings.

| Model | Embed Dim. | MLP Dim. | FiLM | Heads | Blocks | Patch Size | Params |
|---|---|---|---|---|---|---|---|
| Bubbleformer-S | 384 | 1536 | 9 | 6 | 12 | 16 | 29.5 M |
| Bubbleformer-L | 768 | 3072 | 9 | 12 | 12 | 16 | 115.8 M |

Table 4: Architectural specifications of the Bubbleformer models used in our experiments.

Each model is trained in a supervised manner using **teacher forcing** [55] and **temporal bundling** [5]. The model inputs are always the simulation ground truths $[\phi, T, \vec{u}]_{t-k\,:\,t-1}$ and the model learns to predict the next k states, $[\phi, T, \vec{u}]_{t\,:\,t+k-1}$, in a bundled fashion. We also make the following architectural decisions while training:

- **Patch Embedding and Reconstruction** Hierarchical MLP [50].
- **MLP Activation** GeLU [26].
- **Data Normalization** None. We find that a valid signed distance field is essential to learn renucleation. As such, we do not normalize the data and use a relative L2 loss to perform the learning task.
- **FiLM Layer** 9 fluid parameters are used to condition the model. These parameters are as follows: Reynolds Number(Re), relative specific heat capacity($C_p'$), relative viscosity($\mu'$), relative density($\rho'$), relative thermal conductivity($k'$), Stefan Number(St), Prandtl Number(Pr), nucleation wait time and heater temperature.
- **Attention and Feature Scaling** We use attention scaling in both the spatial and temporal attention blocks, but perform feature scaling only once after each spatio-temporal block.

**Hardware.** All models were trained using Distributed Data Parallel on 4 NVIDIA A30 GPUs for the Bubbleformer-S models and 2 NVIDIA A100 GPUs for the Bubbleformer-L models. The models were trained for 250 epochs with a single GPU batch size of 4, which took around 48 hours for the S models and 60 hours for the L models.

### C.2  Hyperparameter Settings

The hyperparameters are tuned for the single bubble validation case and then replicated across all other forecasting scenarios. We found that Lion[7] optimizer performs better than Adam and AdamW in our use case. Following the authors recommendations, we set the learning rate to a lower value and

weight decay to a higher value than what is generally used for AdamW. We do not perform any data augmentations or transforms during training as we find the existing amount of data to be sufficient for training large transformer models.

Table 5: **Training Configuration for Forecasting Experiments.** Summary of hyperparameter settings used during training of Bubbleformer models for boiling trajectory forecasting.

| Hyperparameter | Value |
|---|---|
| Number of Epochs | 250 |
| Iterations per Epoch | 1000 |
| Batch Size | 4 |
| Optimizer | Lion |
| Weight Decay | 0.1 |
| Base Learning Rate | $5 \times 10^{-4}$ |
| Learning Rate Warmup Steps | 1000 |
| Learning Rate Scheduler | Cosine Annealing |
| Minimum Learning Rate | $1 \times 10^{-6}$ |
| History Window Size | 5 |
| Future Forecast Window | 5 |

## C.3 Single Bubble Validation

Interacting bubble dynamics is a stochastic process. To validate Bubbleformer forecasting, we consider a controlled single bubble originating from a nucleation site. The dynamics include nucleation, growth, departure to the next bubble nucleating at the same location after a wait time. We generate 11 single bubble simulations each for two fluids, FC-72 and R515B corresponding to the same wall superheat ($T_{wall} - T_{liquid}$) values $[29, 32, 33, 34, 36, 37, 38, 40, 41, 42, 45]$. The nucleation wait time is set to different values for the fluids, 0.4 simulation time units for FC-72 and 0.6 simulation time units for R515B. The simulations are run for 400 time units and frames are plotted every 0.2 units to generate 2000 frames per simulation. These simulations are then used to train a Bubbleformer-S forecasting model. The training set consists of 12 simulations corresponding to wall superheat values $[32, 34, 36, 38, 40, 42]$ for both fluids, leaving behind 2 out-of-distribution$[29, 45]$ and 3 in-distribution$[33, 37, 41]$ test trajectories.

Upon completion of training, we do an autoregressive rollout for the 10 test trajectories across both fluids for 200 timesteps (around 3 bubble nucleation, growth and departure cycles) and report the physical error metrics for the in-distribution test trajectories in Table 6. The simulation ground truths and the corresponding model forecasts are compared against each other for one bubble cycle in Figures 7 and 8, which shows excellent performance on an in-distribution test trajectory. However, the performance of the model on the out-of-distribution trajectories is significantly worse as seen in Figure 9 a and b, highlighting a potential failure mode for our Bubbleformer models. We also observe that the results are significantly better for R515B compared to FC-72, especially in the later timesteps of the rollout. While the decreasing trend of bubble growth time with increasing wall superheat is captured correctly for R515B, the model fails to do so for FC-72. Moreover, the vapor volume does not maintain a good correlation with the simulation for the latter half of the rollout in FC-72 which results in incorrect artifacts in the temperature field. This explains the high KL divergence for heat flux seen in FC-72 test trajectories.

## C.4 Pool Boiling Results and Discussion

The pool boiling dataset consists of 120 simulations spanning across 3 different fluids (FC-72, R515B and LN2) and two different boiling configurations (Saturated and Subcooled Pool Boiling). The simulations span the entire nucleate boiling region of the boiling curve for the fluids, shown in Tables 14, 15, and 16. The learning task is simplified to only two fluids, FC-72 and R515B of a specific boiling configuration. Thus we train 4 forecasting models, a Bubbleformer-S and a Bubbleformer-L each for Saturated and Subcooled pool boiling of the two fluids.

Table 6: **Forecasting Accuracy: Single Bubble Pool Boiling.** Physics-based error metrics evaluated at in-distribution wall superheats for the Bubbleformer-S model. Results are reported for two fluids (R515B and FC72) at increasing wall superheat conditions. The metrics quantify accuracy in predicting interface geometry (Eikonal Loss), conserving mass (Relative Vapor Volume [RVV] Error), and capturing heat flux distributions (KL Divergence). Lower values indicate better model performance.

| Wall Superheat | R515B | | | FC72 | | |
| --- | --- | --- | --- | --- | --- | --- |
| (°C) | Eikonal Loss | RVV Err | KL Div | Eikonal Loss | RVV Err | KL Div |
| 33 | 0.0755 | 0.1211 | 0.0209 | 0.1468 | 0.2492 | 0.3721 |
| 37 | 0.0757 | 0.0534 | 0.0935 | 0.1815 | 0.1765 | 1.062 |
| 41 | 0.0769 | 0.0279 | 0.1593 | 0.1541 | 0.1332 | 1.044 |

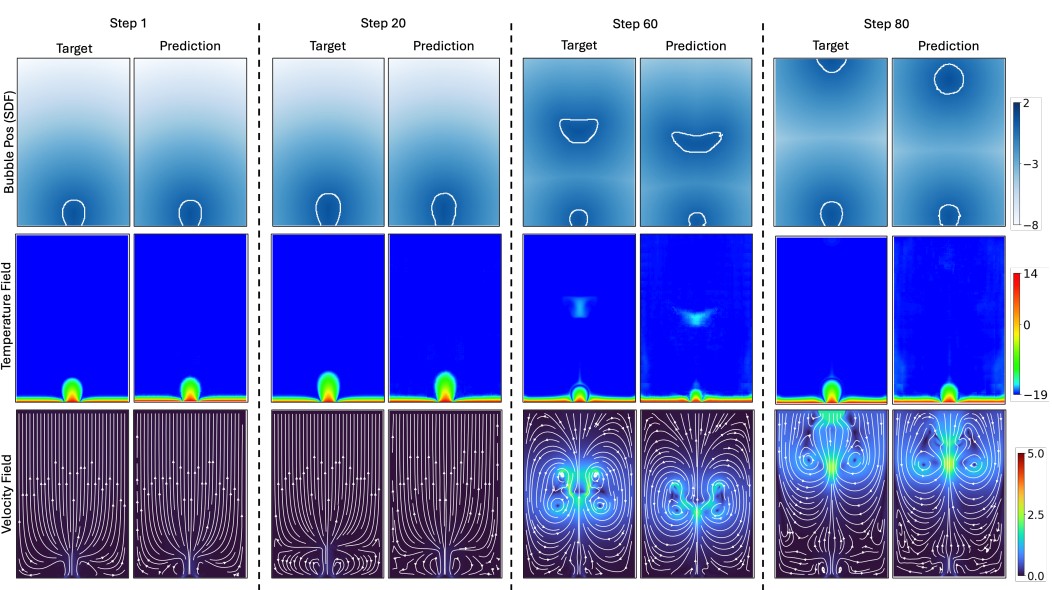

Figure 7: **Single-Bubble Forecasting for R515B at 33 °C Wall Superheat.** Comparison of ground truth and Bubbleformer predicted bubble position (signed distance field), temperature, and velocity fields for an idealized single-bubble scenario. Forecasting was performed using the Bubbleformer-S model under saturated pool boiling conditions.

Owing to the poor out-of-distribution performance in the single-bubble study, we leave out 2 in-distribution test trajectories for each fluid to evaluate our trained models. Interestingly, as reported in Table 7, we observe that both Bubbleformer-S and Bubbleformer-L perform equivalently well on the saturated pool boiling task. However, in Table 8, we observe that for the much harder subcooled pool boiling task, Bubbleformer-L significantly outperforms Bubbleformer-S. We hypothesize that the increased embedding dimension is necessary to represent the richer features such as condensation vortices seen in the subcooled pool boiling study.

## C.5 Flow Boiling Results and Discussion

Flow boiling forecasting models are trained on the data set with varying inlet velocity scales ranging from 1.5 to 2.9. In this case as well, we leave out trajectory-2.2, an in-distribution test case to evaluate our models. In contrast to UNet and FNO models, the patching mechanism of transformers helps the model learn a global context, which is paramount for learning a good flow boiling forecasting model. However, in this case, we observe that the Bubbleformer-S model performs better than the Bubbleformer-L model across all three metrics.

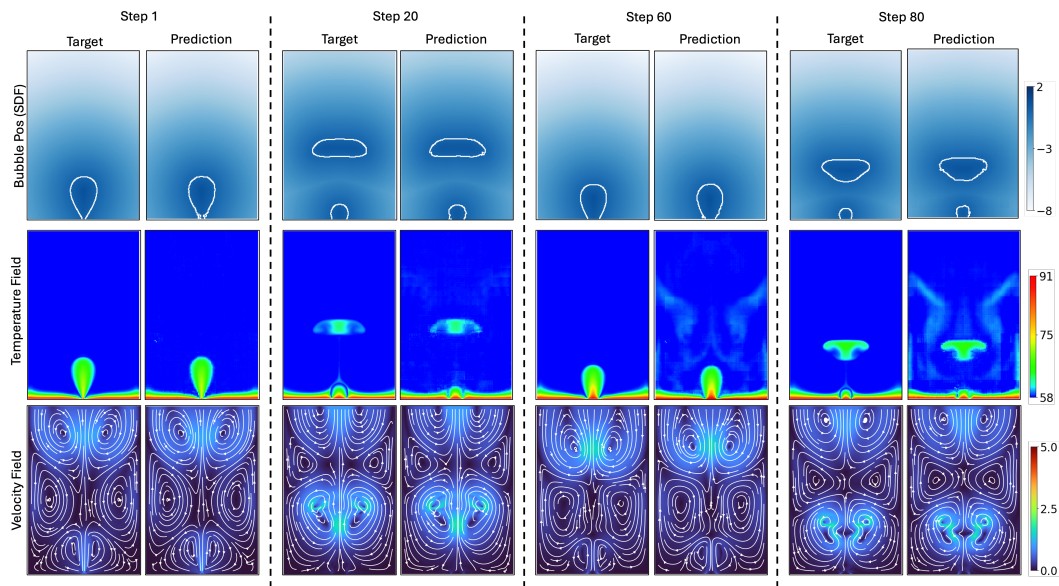

Figure 8: **Single-Bubble Forecasting for FC-72 at 33 °C Wall Superheat.** Comparison of ground truth and Bubbleformer predicted bubble position (signed distance field), temperature, and velocity fields for an idealized single-bubble scenario. Forecasting was performed using the Bubbleformer-S model under saturated pool boiling conditions.

Table 7: **Forecasting Accuracy: Saturated Pool Boiling.** Mean error metrics for Bubbleformer-S and Bubbleformer-L models evaluated across two fluids (FC-72 and R515B) and varying wall superheat. The metrics quantify accuracy in predicting interface geometry (Eikonal Loss), conserving mass (Relative Vapor Volume [RVV] Error), and capturing heat flux distributions (KL Divergence). Bolded values indicate the best (lowest) error per fluid-temperature condition.

| Model | Fluid | Heater Temp | Mean Eikonal Loss | Mean RVV Err | KL Div |
|---|---|---|---|---|---|
| Bubbleformer-S | FC-72 | 91 °C | 0.132 | 0.073 | **0.335** |
| | | 101 °C | **0.150** | 0.082 | **0.277** |
| | R515B | 18 °C | 0.144 | 0.128 | 0.065 |
| | | 28 °C | **0.130** | 0.094 | 0.145 |
| Bubbleformer-L | FC-72 | 91 °C | **0.124** | **0.039** | 0.360 |
| | | 101 °C | 0.157 | **0.042** | 0.318 |
| | R515B | 18 °C | **0.141** | **0.093** | **0.023** |
| | | 28 °C | 0.137 | **0.034** | **0.027** |

Table 8: **Forecasting Accuracy: Subcooled Pool Boiling.** Mean error metrics for Bubbleformer-S and Bubbleformer-L models evaluated across two fluids (FC-72 and R515B) and varying wall superheat. The metrics quantify accuracy in predicting interface geometry (Eikonal Loss), conserving mass (Relative Vapor Volume [RVV] Error), and capturing heat flux distributions (KL Divergence). Bolded values indicate the best (lowest) error per fluid-temperature condition.

| Model | Fluid | Heater Temp | Mean Eikonal Loss | Mean RVV Err | KL Div |
|---|---|---|---|---|---|
| Bubbleformer-S | FC-72 | 97 °C | **0.119** | 2.658 | **1.060** |
| | | 107 °C | 0.166 | 2.318 | 0.722 |
| | R515B | 30 °C | **0.136** | 3.656 | 0.154 |
| | | 40 °C | **0.157** | 2.772 | 0.554 |
| Bubbleformer-L | FC-72 | 97 °C | 0.218 | **0.376** | 1.407 |
| | | 107 °C | **0.163** | **0.123** | **0.124** |
| | R515B | 30 °C | 0.155 | **0.078** | **0.048** |
| | | 40 °C | 0.190 | **0.056** | **0.049** |

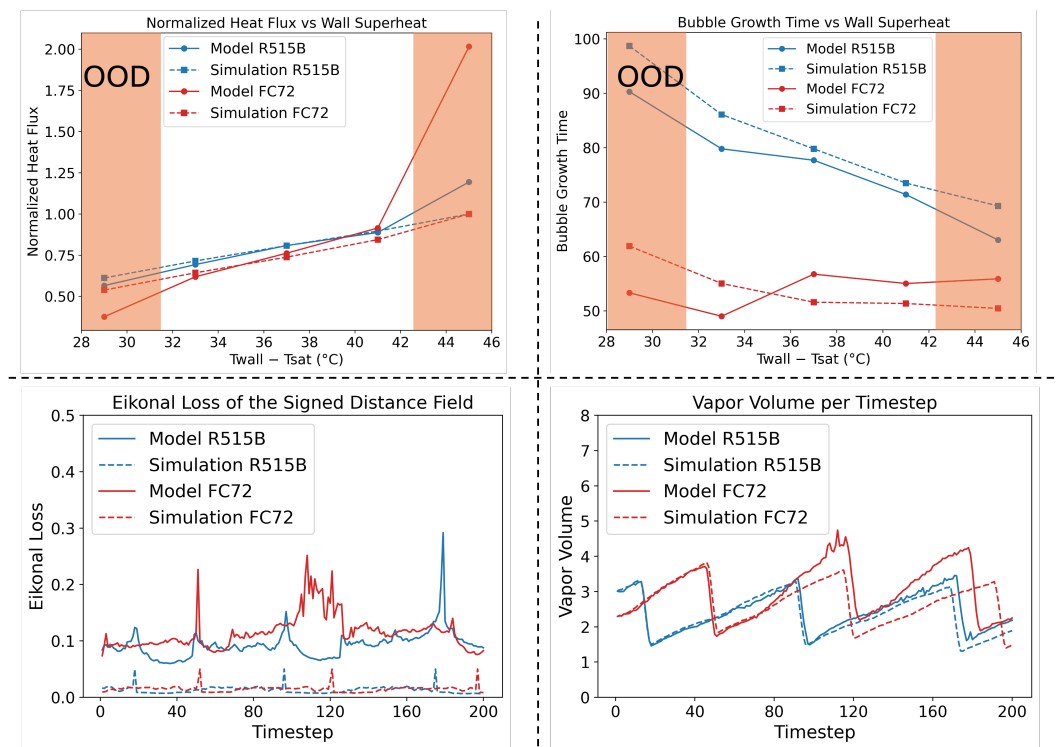

Figure 9: **Forecasting Performance: Single Bubble Pool Boiling.** Metrics shown across varying wall superheats and timesteps for R515B and FC-72 under saturated pool boiling conditions. We report bubble growth time(ms), heat flux prediction (system-level quantity), Eikonal loss (interface geometry), and vapor volume (mass conservation) to assess the physical plausibility of forecasts. Results highlight the model's robustness across fluid types and close alignment with ground truth simulations.

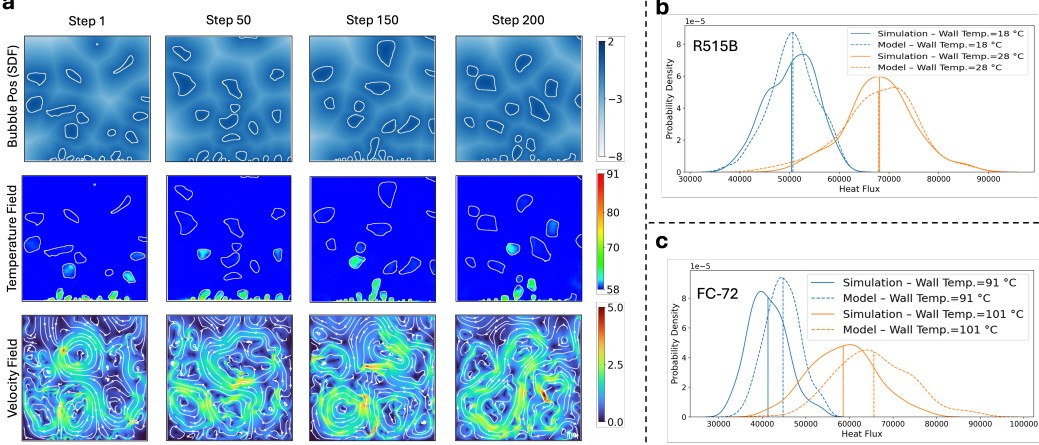

Figure 10: **Forecasting Results: Saturated Pool Boiling**. (a) Rollout for a Bubbleformer-L model on an unseen pool boiling trajectory, FC-72 at heater temperature= 91 °C. (b) and (c) Comparison of predicted vs. ground-truth heat flux PDFs for R515B and FC-72 respectively at different wall temperatures.

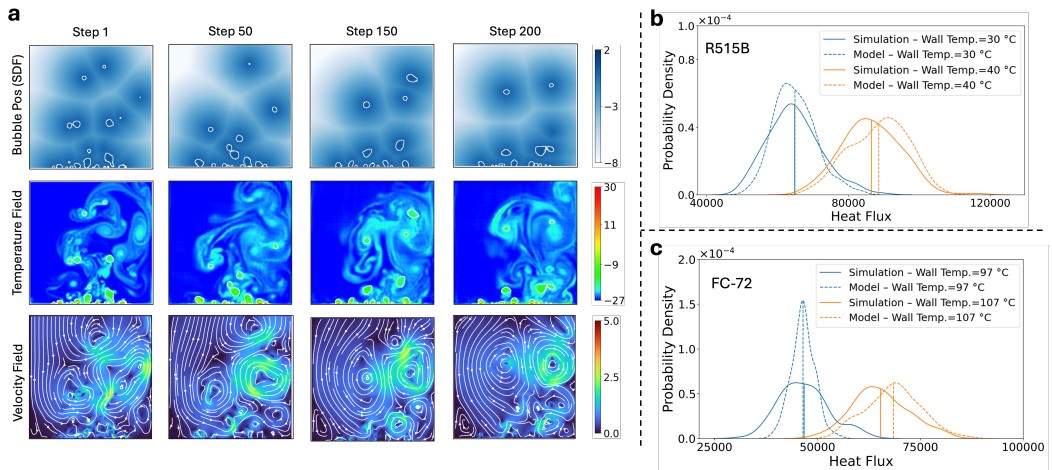

Figure 11: **Forecasting Results: Subcooled Pool Boiling**. (a) Rollout for a Bubbleformer-L model on an unseen pool boiling trajectory, R515B at heater temperature= 30 °C. (b) and (c) Comparison of predicted vs. ground-truth heat flux PDFs for R515B and FC-72 respectively at different wall temperatures.

Table 9: **Forecasting Accuracy: Flow Boiling.** Mean error metrics for Bubbleformer-S and Bubbleformer-L on a flow boiling test trajectory with inlet velocity = 2.2 m/s using FC-72 as working fluid. Metrics reflect the model's ability to predict interface geometry (Eikonal Loss), conserve mass (RVV Error), and system-level heat flux distribution (KL Divergence). Lower values indicate better performance.

| Model | Fluid | Inlet Velocity | Mean Eikonal Loss | Mean RVV Err. | KL Div. |
|---|---|---|---|---|---|
| Bubbleformer-S | FC-72 | 2.2 | **0.107** | **0.105** | **0.032** |
| Bubbleformer-L | FC-72 | 2.2 | 0.203 | 0.124 | 0.060 |

## C.6 Computational Efficiency

We observe that Bubbleformer is more efficient compared to the baseline models in the paper (UNet and FFNO). In Table 10 we report the number of parameters and the inference time per rollout timestep on a single NVIDIA A30 GPU.

Table 10: Comparison of model sizes and inference speeds per rollout step.

| Model | Parameters (M) | Time per rollout step (s) |
|---|---|---|
| Bubbleformer-S | 29.5 | 0.016 |
| Bubbleformer-L | 115.8 | 0.028 |
| FFNO | 119.3 | 0.043 |
| Modern UNet | 566.7 | 0.032 |

The efficiency gain is mainly due to axial attention which reduces the complexity from $\mathcal{O}(HWT)^2$ in joint space-time attention to $\mathcal{O}(H^2 + W^2 + T^2)$. It is also important to note that the number of Fourier modes in the low pass filter in FFNO architectures largely determines its performance. Literature[21] dictates that the number of Fourier modes should be closer to two-thirds of the spatial domain resolution to preserve spectral information while avoiding aliasing errors. This effectively places a constraint on the number of parameters in the model to input resolution. The modern UNet architecture we chose is the one publicly released as part of BubbleML[25] benchmarks, and the effect on accuracy and inference time by reducing the number of parameters remains to be explored.

# D   Prediction

## D.1   Benchmark Models

Aside from our bubbleformer model, we evaluate two baseline neural PDE solvers used in the BubbleML[25] benchmark: **UNet$_{\mathbf{mod}}$** and **F-FNO**.

1. **UNet$_{\mathbf{mod}}$**: UNet is a commonly used image-to-image architecture in computer vision tasks such as image segmentation. While standard UNet models are not specifically designed for PDE learning—especially when training data come from numerical simulations with varying spatial resolutions—modern adaptations of UNet remain effective in several benchmarks [24, 40]. UNet$_{mod}$ is a variant of UNet that incorporates wide residual connections and group normalization. It is used here as a general-purpose baseline for learning PDE dynamics.

2. **F-FNO**: F-FNO (Factorized Fourier Neural Operator) [51] is a type of neural operator designed to efficiently solve PDEs by learning mappings between function spaces. Neural operators aim to approximate solution operators of PDEs. Given an initial condition $u_0$, a neural operator is defined as a mapping $\mathcal{M} : [0, t_{\max}] \times \mathcal{A} \to \mathcal{A}$, where $\mathcal{A}$ is an infinite-dimensional function space, and $\mathcal{M}(t, u_0) = u_t$ [38, 33]. In practice, training such models requires a large set of initial conditions and their corresponding simulation trajectories, which is computationally expensive for datasets like BubbleML. To mitigate this, we adopt an autoregressive training setup, which has also been used in prior work [40, 42, 5]. F-FNO improves the scalability of Fourier Neural Operators by factorizing the Fourier transform across spatial dimensions. This reduces the parameter count per Fourier weight matrix to $\mathcal{O}(H^2 MD)$, enabling the use of more Fourier modes or deeper model architectures. It also introduces a modified residual structure where the residual connection is applied after the nonlinearity.

## D.2   Evaluation Metrics

To quantitatively assess model performance, we use a set of error metrics computed over spatiotemporal fields. Let $\hat{\mathbf{y}}_t \in \mathbb{R}^{H \times W}$ denote the predicted field at time step $t$, and let $\mathbf{y}_t \in \mathbb{R}^{H \times W}$ be the corresponding ground truth.

**Root Mean Square Error (RMSE).** RMSE measures the average magnitude of the prediction error:

$$\text{RMSE} = \frac{1}{T} \sum_{t=1}^{T} \sqrt{\frac{1}{HW} \|\hat{\mathbf{y}}_t - \mathbf{y}_t\|_2^2}, \tag{21}$$

where $T$ is the number of time steps and $H \times W$ denotes the spatial resolution.

**Relative L2 Error.** This metric evaluates the normalized L2 distance between prediction and ground truth:

$$\text{Relative L2 Error} = \frac{1}{T} \sum_{t=1}^{T} \frac{\|\hat{\mathbf{y}}_t - \mathbf{y}_t\|_2}{\|\mathbf{y}_t\|_2 + \varepsilon}, \tag{22}$$

where $\varepsilon$ is a small constant added for numerical stability.

**Max Relative L2 Error.** We also report the worst-case frame-wise relative error:

$$\text{Max Relative L2 Error} = \max_{1 \leq t \leq T} \frac{\|\hat{\mathbf{y}}_t - \mathbf{y}_t\|_2}{\|\mathbf{y}_t\|_2 + \varepsilon}. \tag{23}$$

**Maximum Error.** This measures the largest pointwise squared error across all spatial and temporal locations:

$$\text{Max Error} = \max_{t,i,j} \left( \hat{y}_{t,i,j} - y_{t,i,j} \right)^2. \tag{24}$$

**Interface RMSE.** We evaluate error specifically on interface regions identified by a signed distance function:

$$\text{Interface RMSE} = \sqrt{\frac{1}{|\mathcal{I}|} \sum_{(i,j) \in \mathcal{I}} \left( \hat{y}_{i,j} - y_{i,j} \right)^2}, \tag{25}$$

where $\mathcal{I}$ is the set of interface pixel indices.

**Boundary RMSE (BRMSE).** To assess prediction accuracy near domain edges, we compute RMSE using only boundary values. The boundary is extracted by concatenating the values from all four edges (left, right, top, and bottom) of each frame:

$$\text{BRMSE} = \sqrt{\frac{1}{|\mathcal{B}|} \sum_{(i,j) \in \mathcal{B}} \left( \hat{y}_{i,j} - y_{i,j} \right)^2}, \tag{26}$$

where $\mathcal{B}$ is the set of all boundary pixels over the entire temporal rollout.

**Fourier Spectrum Error.** To quantify the frequency-dependent discrepancy between predicted and true fields, we compute radial averages of the squared differences in the spatial Fourier domain.

Let $\hat{y}_t, y_t \in \mathbb{R}^{H \times W}$ denote the predicted and true fields at time $t$, and let $\hat{Y}_t = \mathcal{F}[\hat{y}_t]$, $Y_t = \mathcal{F}[y_t]$ be their respective 2D discrete Fourier transforms. Define the Fourier error as:

$$E_t(i,j) = \left| \hat{Y}_t(i,j) - Y_t(i,j) \right|^2, \tag{27}$$

where $(i,j)$ indexes spatial frequencies.

We convert Cartesian frequency coordinates to radial bins via:

$$k = \left\lfloor \sqrt{i^2 + j^2} \right\rfloor, \tag{28}$$

and compute the radially averaged spectrum:

$$\bar{E}_t(k) = \sum_{\sqrt{i^2+j^2} \approx k} E_t(i,j). \tag{29}$$

We then average across all timesteps:

$$\bar{E}(k) = \frac{1}{T} \sum_{t=1}^{T} \bar{E}_t(k), \tag{30}$$

normalize by domain size, and report three band-aggregated errors:

- **Low-frequency error:** $\dfrac{1}{k_{\text{low}}} \displaystyle\sum_{k=0}^{k_{\text{low}}-1} \bar{E}(k)$, with $k_{\text{low}} = 4$

- **Mid-frequency error:** $\dfrac{1}{k_{\text{mid}}} \displaystyle\sum_{k=k_{\text{low}}}^{k_{\text{high}}-1} \bar{E}(k)$, with $k_{\text{mid}} = 8$ and $k_{\text{high}} = 12$

- **High-frequency error:** $\dfrac{1}{K - k_{\text{high}}} \displaystyle\sum_{k=k_{\text{high}}}^{K-1} \bar{E}(k)$, where $K = \min(H/2, W/2)$

This metric captures model fidelity across scales: low-$k$ (large structures), mid-$k$ (medium textures), and high-$k$ (fine-grained details). The errors are scaled by the physical domain size $(L_x, L_y)$ to maintain consistency across resolutions.

## D.3 Hyperparameter Settings

The hyperparameter settings are the same as those used in the forecasting task shown in Table 5. The settings for UNet$_{\text{mod}}$ and F-FNO follow those reported in the BubbleML [25] (Appendix C.3, Table 7).

## D.4 Additional Results and Discussion

We present the results for three boiling scenarios in this section: Subcooled Pool Boiling with FC-72 in Table 11 and Figure 5, Saturated Pool Boiling with R-515B in Table 12 and Figure 12, and Inlet Velocity Flow Boiling with FC-72 in Table 13 and Figure 13. For all prediction experiments, we perform rollouts over 800 timesteps.

Additionally, we introduce a max relative L2 error metric, which captures the worst-case mean relative L2 error across the rollout window. This metric highlights the model's robustness under compounding prediction error.

Bubbleformer outperforms UNet$_{\text{mod}}$ and F-FNO across most metrics in Table 11, as discussed in Section 6.3. Since UNet$_{\text{mod}}$ and F-FNO failed to produce valid results for the Flow Boiling cases, we excluded them from Table 13. Additionally, due to compute and time constraints, we did not evaluate these baselines on the Saturated Pool Boiling scenario in Table 12. Nevertheless, we expect Bubbleformer to maintain superior performance, as its inherent ability to resolve both sharp, non-smooth interfaces and long-range dependencies gives it a clear advantage in boiling flow simulations.

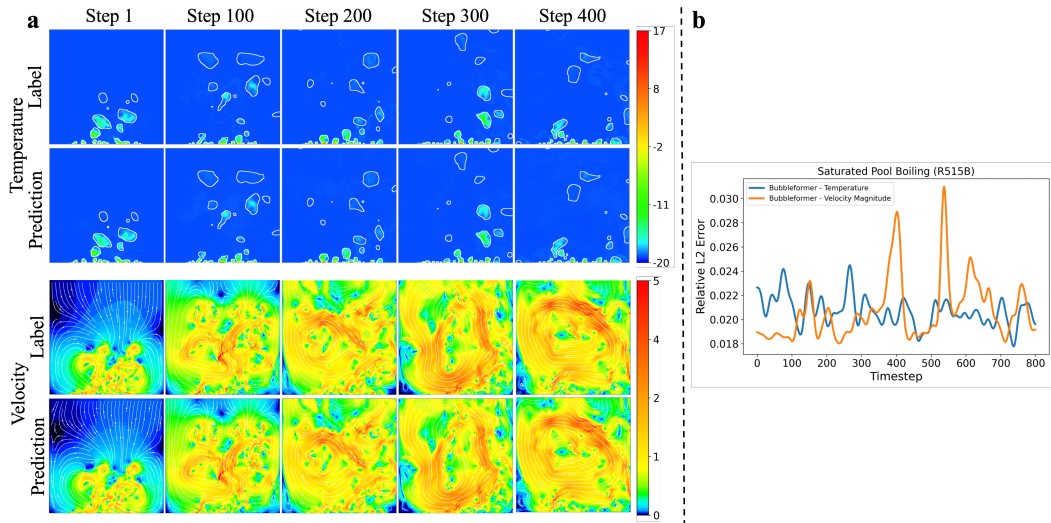

Figure 12: **Saturated Pool Boiling Prediction.** (a) Predicted temperature and velocity fields from the Bubbleformer-S model on an unseen subcooled pool boiling trajectory for R515B. (b) Relative L2 error over 800 rollout timesteps for temperature and velocity magnitude (combined x and y components) of Bubbleformer-S.

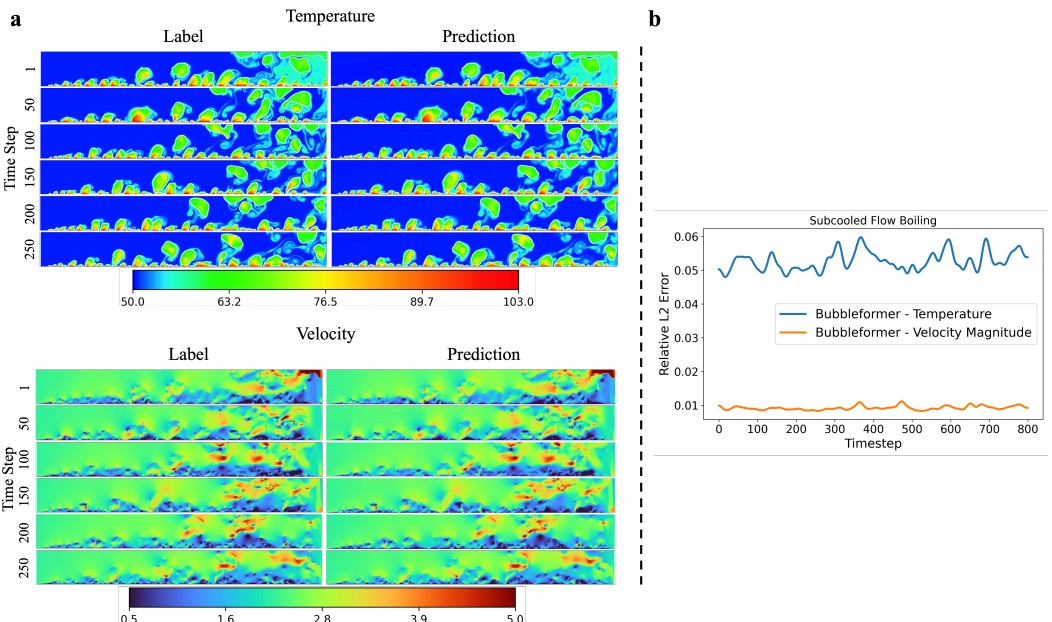

Figure 13: **Flow Boiling Inlet Velocity Prediction.** (a) Predicted temperature and velocity fields from the Bubbleformer-S model on an unseen flow boiling trajectory for FC-72. (b) Relative L2 error over 800 rollout timesteps for temperature and velocity magnitude (combined x and y components) of Bubbleformer-S.

| Category | Metric | Bubbleformer | FFNO | UNet$_{mod}$ |
|---|---|---|---|---|
| **Temperature** | Rel L2 | **0.031048** | 0.089853 | 0.096906 |
| | RMSE | **0.029122** | 0.086781 | 0.093600 |
| | BRMSE | **0.109284** | 0.247217 | 0.276180 |
| | IRMSE | **0.126722** | 0.210089 | 0.181712 |
| | MaxErr | 3.675776 | **2.488442** | 2.790049 |
| | Max L2 | **0.075884** | 0.127008 | 0.133258 |
| | Fourier Low | **0.180419** | 0.938612 | 0.882603 |
| | Fourier Mid | **0.153419** | 0.639546 | 0.758667 |
| | Fourier High | **0.051429** | 0.085806 | 0.096905 |
| **Velocity X** | Rel L2 | **0.173494** | 0.806996 | 0.933271 |
| | RMSE | **0.008408** | 0.021742 | 0.024675 |
| | BRMSE | 0.030632 | 0.023827 | **0.023260** |
| | IRMSE | 0.059506 | **0.050735** | 0.051402 |
| | MaxErr | 0.785328 | **0.347980** | 0.369229 |
| | Max L2 | **0.405068** | 1.326638 | 1.506975 |
| | Fourier Low | **0.049933** | 0.340134 | 0.417801 |
| | Fourier Mid | **0.042705** | 0.124967 | 0.125568 |
| | Fourier High | 0.015219 | 0.011966 | **0.011803** |
| **Velocity Y** | Rel L2 | **0.026967** | 0.681400 | 0.765296 |
| | RMSE | **0.008006** | 0.024380 | 0.026884 |
| | BRMSE | **0.012534** | 0.016269 | 0.018310 |
| | IRMSE | 0.051713 | 0.048940 | **0.047879** |
| | MaxErr | 1.699735 | **0.791652** | 0.843999 |
| | Max L2 | **0.062572** | 1.247504 | 1.315714 |
| | Fourier Low | **0.059147** | 0.346206 | 0.373078 |
| | Fourier Mid | **0.043992** | 0.134964 | 0.142891 |
| | Fourier High | 0.013107 | **0.010859** | 0.011118 |

Table 11: **Subcooled Pool Boiling (FC-72) Velocity and Temperature Prediction Metrics.** Metrics include Relative L2 error, RMSE, BRMSE, Max Error, Maximum L2 error, and Fourier Errors for each physical field.

| Category | Metric | Bubbleformer |
|---|---|---|
| **Temperature** | Rel L2 | 0.020703 |
| | RMSE | 0.018814 |
| | BRMSE | 0.121156 |
| | MaxErr | 3.107522 |
| | Max L2 | 0.040473 |
| | Fourier Low | 0.023166 |
| | Fourier Mid | 0.041523 |
| | Fourier High | 0.033939 |
| **Velocity X** | Rel L2 | 0.099593 |
| | RMSE | 0.004759 |
| | BRMSE | 0.009329 |
| | IRMSE | 0.016655 |
| | MaxErr | 1.969497 |
| | Max L2 | 0.310537 |
| | Fourier Low | 0.018420 |
| | Fourier Mid | 0.016868 |
| | Fourier High | 0.007180 |
| **Velocity Y** | Rel L2 | 0.021445 |
| | RMSE | 0.004838 |
| | BRMSE | 0.004315 |
| | IRMSE | 0.016664 |
| | MaxErr | 0.674551 |
| | Max L2 | 0.057292 |
| | Fourier Low | 0.019422 |
| | Fourier Mid | 0.017947 |
| | Fourier High | 0.007236 |

Table 12: **Saturated Pool Boiling (R515B) Prediction Metrics**

| Category | Metric | Bubbleformer |
|---|---|---|
| **Temperature** | Rel L2 | 0.052712 |
| | RMSE | 0.046211 |
| | BRMSE | 0.178124 |
| | IRMSE | 0.121292 |
| | MaxErr | 3.139554 |
| | Max L2 | 0.110760 |
| | Fourier Low | 0.070464 |
| | Fourier Mid | 0.120353 |
| | Fourier High | 0.146381 |
| **Velocity X** | Rel L2 | 0.009240 |
| | RMSE | 0.004052 |
| | BRMSE | 0.013071 |
| | IRMSE | 0.012441 |
| | MaxErr | 0.432624 |
| | Max L2 | 0.029023 |
| | Fourier Low | 0.026810 |
| | Fourier Mid | 0.018300 |
| | Fourier High | 0.011747 |
| **Velocity Y** | Rel L2 | 0.306108 |
| | RMSE | 0.005391 |
| | BRMSE | 0.008027 |
| | IRMSE | 0.017001 |
| | MaxErr | 0.658709 |
| | Max L2 | 0.598994 |
| | Fourier Low | 0.011650 |
| | Fourier Mid | 0.014031 |
| | Fourier High | 0.016585 |

Table 13: **Flow Boiling Inlet Velocity (FC-72) Prediction Metrics.**

# E   Datasheet

| Study | Fluid (T_bulk) | Wall Temp. | Nuc. Sites | T_wall - T_bulk | Sat. Temp. | Stefan Num. |
|---|---|---|---|---|---|---|
| Saturated | FC-72 (58°C) | 75 | 5 | 17 | 0 | 0.2219 |
| | | 76 | 6 | 18 | 0 | 0.2349 |
| | | 78 | 8 | 20 | 0 | 0.2610 |
| | | 80 | 10 | 22 | 0 | 0.2871 |
| | | 82 | 12 | 24 | 0 | 0.3132 |
| | | 84 | 14 | 26 | 0 | 0.3393 |
| | | 86 | 16 | 28 | 0 | 0.3654 |
| | | 88 | 18 | 30 | 0 | 0.3915 |
| | | 90 | 20 | 32 | 0 | 0.4176 |
| | | 91 | 21 | 33 | 0 | 0.4307 |
| | | 92 | 22 | 34 | 0 | 0.4437 |
| | | 94 | 24 | 36 | 0 | 0.4698 |
| | | 96 | 26 | 38 | 0 | 0.4959 |
| | | 98 | 28 | 40 | 0 | 0.5220 |
| | | 100 | 30 | 42 | 0 | 0.5481 |
| | | 101 | 31 | 43 | 0 | 0.5612 |
| | | 102 | 32 | 44 | 0 | 0.5742 |
| | | 104 | 34 | 46 | 0 | 0.6003 |
| | | 106 | 36 | 48 | 0 | 0.6264 |
| | | 107 | 37 | 49 | 0 | 0.6395 |
| Subcooled | FC-72 (50°C) | 85 | 3 | 35 | 0.2286 | 0.4568 |
| | | 86 | 4 | 36 | 0.2222 | 0.4698 |
| | | 88 | 6 | 38 | 0.2105 | 0.4959 |
| | | 90 | 8 | 40 | 0.2000 | 0.5220 |
| | | 92 | 10 | 42 | 0.1905 | 0.5481 |
| | | 94 | 12 | 44 | 0.1818 | 0.5742 |
| | | 96 | 14 | 46 | 0.1739 | 0.6003 |
| | | 97 | 15 | 47 | 0.1702 | 0.6134 |
| | | 98 | 16 | 48 | 0.1667 | 0.6264 |
| | | 100 | 18 | 50 | 0.1600 | 0.6525 |
| | | 102 | 20 | 52 | 0.1538 | 0.6786 |
| | | 104 | 22 | 54 | 0.1481 | 0.7047 |
| | | 106 | 24 | 56 | 0.1429 | 0.7308 |
| | | 107 | 25 | 57 | 0.1404 | 0.7439 |
| | | 108 | 26 | 58 | 0.1379 | 0.7569 |
| | | 110 | 28 | 60 | 0.1333 | 0.7830 |
| | | 112 | 30 | 62 | 0.1290 | 0.8091 |
| | | 114 | 32 | 64 | 0.1250 | 0.8352 |
| | | 116 | 34 | 66 | 0.1212 | 0.8613 |
| | | 117 | 35 | 67 | 0.1194 | 0.8744 |

Table 14: Boiling Curve Data for FC-72

| Study | Fluid (T_bulk) | Wall Temp. | Nuc. Sites | T_wall - T_bulk | Sat. Temp. | Stefan Num. |
|-------|----------------|------------|------------|-----------------|------------|-------------|
| Saturated | R515B (-19°C) | 4 | 5 | 23 | 0 | 0.1525 |
| | | 5 | 6 | 24 | 0 | 0.1591 |
| | | 7 | 8 | 26 | 0 | 0.1724 |
| | | 9 | 10 | 28 | 0 | 0.1857 |
| | | 11 | 12 | 30 | 0 | 0.1989 |
| | | 13 | 14 | 32 | 0 | 0.2122 |
| | | 15 | 16 | 34 | 0 | 0.2255 |
| | | 17 | 18 | 36 | 0 | 0.2387 |
| | | 18 | 19 | 37 | 0 | 0.2453 |
| | | 19 | 20 | 38 | 0 | 0.2520 |
| | | 21 | 22 | 40 | 0 | 0.2652 |
| | | 23 | 24 | 42 | 0 | 0.2785 |
| | | 25 | 26 | 44 | 0 | 0.2918 |
| | | 27 | 28 | 46 | 0 | 0.3050 |
| | | 28 | 29 | 47 | 0 | 0.3117 |
| | | 29 | 30 | 48 | 0 | 0.3183 |
| | | 31 | 32 | 50 | 0 | 0.3316 |
| | | 33 | 34 | 52 | 0 | 0.3448 |
| | | 35 | 36 | 54 | 0 | 0.3581 |
| | | 36 | 37 | 55 | 0 | 0.3647 |
| Subcooled | R515B (-27°C) | 14 | 3 | 41 | 0.1951 | 0.2719 |
| | | 15 | 4 | 42 | 0.1905 | 0.2785 |
| | | 17 | 6 | 44 | 0.1818 | 0.2918 |
| | | 19 | 8 | 46 | 0.1739 | 0.3050 |
| | | 21 | 10 | 48 | 0.1667 | 0.3183 |
| | | 23 | 12 | 50 | 0.1600 | 0.3316 |
| | | 25 | 14 | 52 | 0.1538 | 0.3448 |
| | | 27 | 16 | 54 | 0.1481 | 0.3581 |
| | | 29 | 18 | 56 | 0.1429 | 0.3713 |
| | | 30 | 19 | 57 | 0.1404 | 0.3780 |
| | | 31 | 20 | 58 | 0.1379 | 0.3846 |
| | | 33 | 22 | 60 | 0.1333 | 0.3979 |
| | | 35 | 24 | 62 | 0.1290 | 0.4111 |
| | | 37 | 26 | 64 | 0.1250 | 0.4244 |
| | | 39 | 28 | 66 | 0.1212 | 0.4376 |
| | | 40 | 29 | 67 | 0.1194 | 0.4443 |
| | | 41 | 30 | 68 | 0.1176 | 0.4509 |
| | | 43 | 32 | 70 | 0.1143 | 0.4642 |
| | | 45 | 34 | 72 | 0.1111 | 0.4774 |
| | | 46 | 35 | 73 | 0.1096 | 0.4841 |

Table 15: Boiling Curve Data for R515B

| Study | Fluid (T_bulk) | Wall Temp. | Nuc. Sites | T_wall - T_bulk | Sat. Temp. | Stefan Num. |
|---|---|---|---|---|---|---|
| Saturated | LN2 (-196°C) | -191 | 5 | 5 | 0 | 0.0512 |
| | | -190 | 6 | 6 | 0 | 0.0614 |
| | | -188 | 8 | 8 | 0 | 0.0818 |
| | | -186 | 10 | 10 | 0 | 0.1023 |
| | | -184 | 12 | 12 | 0 | 0.1228 |
| | | -182 | 14 | 14 | 0 | 0.1432 |
| | | -180 | 16 | 16 | 0 | 0.1637 |
| | | -178 | 18 | 18 | 0 | 0.1841 |
| | | -176 | 20 | 20 | 0 | 0.2046 |
| | | -175 | 21 | 21 | 0 | 0.2148 |
| | | -174 | 22 | 22 | 0 | 0.2251 |
| | | -172 | 24 | 24 | 0 | 0.2455 |
| | | -170 | 26 | 26 | 0 | 0.2660 |
| | | -168 | 28 | 28 | 0 | 0.2864 |
| | | -166 | 30 | 30 | 0 | 0.3069 |
| | | -165 | 31 | 31 | 0 | 0.3171 |
| | | -164 | 32 | 32 | 0 | 0.3274 |
| | | -162 | 34 | 34 | 0 | 0.3478 |
| | | -160 | 36 | 36 | 0 | 0.3683 |
| | | -159 | 37 | 37 | 0 | 0.3785 |
| Subcooled | LN2 (-204°C) | -181 | 3 | 23 | 0.3478 | 0.2353 |
| | | -180 | 4 | 24 | 0.3333 | 0.2455 |
| | | -178 | 6 | 26 | 0.3077 | 0.2660 |
| | | -176 | 8 | 28 | 0.2857 | 0.2864 |
| | | -174 | 10 | 30 | 0.2667 | 0.3069 |
| | | -172 | 12 | 32 | 0.2500 | 0.3274 |
| | | -170 | 14 | 34 | 0.2353 | 0.3478 |
| | | -168 | 16 | 36 | 0.2222 | 0.3683 |
| | | -166 | 18 | 38 | 0.2105 | 0.3887 |
| | | -165 | 19 | 39 | 0.2051 | 0.3990 |
| | | -164 | 20 | 40 | 0.2000 | 0.4092 |
| | | -162 | 22 | 42 | 0.1905 | 0.4297 |
| | | -160 | 24 | 44 | 0.1818 | 0.4501 |
| | | -158 | 26 | 46 | 0.1739 | 0.4706 |
| | | -156 | 28 | 48 | 0.1667 | 0.4910 |
| | | -155 | 29 | 49 | 0.1633 | 0.5013 |
| | | -154 | 30 | 50 | 0.1600 | 0.5115 |
| | | -152 | 32 | 52 | 0.1538 | 0.5320 |
| | | -150 | 34 | 54 | 0.1481 | 0.5524 |
| | | -149 | 35 | 55 | 0.1455 | 0.5627 |

Table 16: Boiling Curve Data for LN2