# OpenReview forum: "Bubbleformer: Forecasting Boiling with Transformers"
_NeurIPS.cc/2025/Datasets_and_Benchmarks_Track — NeurIPS 2025 Datasets and Benchmarks Track spotlight_

### Official Review · Reviewer_ozA9 · 2025-06-30

**Rating:** 5
**Confidence:** 3

**Summary:**

This paper introduces Bubbleformer, a novel transformer-based spatiotemporal model designed to autonomously forecast chaotic two-phase boiling dynamics. Unlike prior machine learning surrogates that require future simulation inputs (like bubble positions) during inference, . Concurrently, they release BubbleML 2.0, an expanded high-fidelity dataset encompassing a broader range of working fluids, boiling configurations, and flow regimes. A benchmark with various models is provided with this dataset. While datasets and benchmarks are provided with this paper, the main focus of the paper seems to be on their new transformer-based modeling approach -- which is not the main focus of the datasets and benchmark track. Due to this slightly out-of-focus presentation, I recommend a weak accept.

**Dataset Code Accessibility:**

Yes

**Dataset Code Comments:**

Dataset available of HF/Kaggle with dataloading code provided.

**Ethical Considerations:**

No, there are no or only very minor ethics concerns

**Final Justification:**

Valid rebuttal -- raising to 5.

**Limitations Weaknesses:**

The main focus of the paper seems to be on their new transformer-based modeling approach -- which is not the main focus of the datasets and benchmark track.Can the authors comment on the suitability of this presentation approach for the D&B track?

**Strengths Contributions:**

1. High Impact for the Community: The ability to autonomously forecast chaotic boiling dynamics is a relevant real-world application of scientific ML and has direct implications for real-time forecasting, parametric studies, and design optimization in critical applications like thermal management for HPC, AI workloads, and potentially nuclear reactors.
2. Comprehensive datasets with 160 simulations
3. Dataset is available in HF and Kaggle with dataloading code provided.
4. Comprehensive benchmark with good metrics.

---

> ### Author Rebuttal · Authors · 2025-07-31
>
> We thank the reviewer for recognizing the high community impact and comprehensive nature of our contribution. We address the track suitability concern below.
>
> **Datasets and Benchmarks track alignment.** Our submission directly aligns with this track through two primary contributions:
>
> 1. **A substantial dataset advancement:** BubbleML 2.0 represents a significant expansion of multiphase flow datasets in both *scale* (160 high-fidelity simulations) and *diversity* (first dataset to span multiple working fluids, flow regimes, and heater configurations). To our knowledge, no existing open dataset provides this level of physical diversity with accompanying velocity, temperature, and phase fields.
>
> 2. **New tasks and benchmarks:** We introduce and benchmark new learning tasks central to boiling physics, specifically nucleation modeling and autonomous forecasting. Bubbleformer provides a strong baseline for this task, and our benchmark includes multiple prior methods with consistent physically meaningful metrics.
>
> In addition, we adhere to the reproducibility goals of this track: all data, preprocessing, code, and model checkpoints are openly released on HuggingFace and GitHub, with plug-and-play dataloaders. In summary, while we introduce a model to demonstrate the utility of our benchmark, the dataset and tasks are also primary contributions. Therefore, we chose this track for our submission.

---

> > ### Comment · Reviewer_ozA9 · 2025-08-02
> >
> > Valid rebuttal -- raising from 4 to 5

---

### Official Review · Reviewer_if6d · 2025-06-30

**Rating:** 5
**Confidence:** 5

**Summary:**

This large‑scale, multiscale bubble‑multiphase dataset captures the inherently chaotic interactions of gas–liquid interfaces across orders of magnitude in length and time. By providing high‑fidelity, multiphysics simulations under a wide range of operating conditions, it delivers the statistical richness needed for training and validating machine‑learning models of interface dynamics, coalescence, and breakup. The inclusion of pixel‑perfect segmentation masks and precomputed meshes also supports downstream tasks such as automated detection, adaptive mesh generation, and reduced‑order modeling. By overcoming the traditional small‑sample bottleneck in multiphase research, this comprehensive resource accelerates the development of data‑driven tools for reactor design, environmental flow prediction, and process optimization.

**Additional Feedback:**

AMR can be simulated and put it on the fine mesh to release, am I right?

**Dataset Code Accessibility:**

Yes

**Dataset Code Comments:**

To maximize reproducibility and foster broad community use, the authors have chosen permissive open‐source licenses and provided multiple convenient access routes for both code and data.

**Ethical Considerations:**

No, there are no or only very minor ethics concerns

**Final Justification:**

I support data publication.

**Limitations Weaknesses:**

The current Bubbleformer architecture cannot natively operate on AMR grids, necessary for simulating fluids such as water or large real‑world domains. Interpolation to uniform grids can introduce numerical errors that can make model training unstable. But the physics is fully captured as a dataset I think.

**Strengths Contributions:**

The paper delivers a substantive advance—both a novel transformer tailored to multiphase chaos and a comprehensive dataset/metric suite. Its impact will extend to thermal management, energy systems, and the broader surrogate‑modeling community.

---

> ### Author Rebuttal · Authors · 2025-07-31
>
> We thank the reviewer for the positive assessment and insightful question about AMR grids.
>
> As the reviewer notes, BubbleML 2.0 fully captures the underlying physics, providing a solid foundation for future AMR-native architectures. Currently, we interpolate the AMR simulation data to uniform fine grids for Bubbleformer input. However, standard interpolation techniques, such as bilinear and bicubic interpolation, are not divergence-free and introduce continuity (i.e., mass conservation) errors at the liquid-vapor interface. Physics is still captured (the continuity loss residual is ~$10^{-2}$) but this is a source of error for the model during training. We aim to resolve this limitation using architectural improvements that can ingest native AMR grids or through divergence-free interpolation techniques [1] in future work.
>
> [1] A direct-forcing embedded-boundary method with adaptive mesh refinement for fluid–structure interaction problems, Vanella et al, Journal of Computational Physics Vol 229.

---

> > ### Comment · Reviewer_8fgs · 2025-08-09
> >
> > Do the authors intend to say that the bilinear and bicubic interpolations introduce discontinuity instead of continuity? I believe it should be discontinuity at the interface.

---

### Official Review · Reviewer_X3Qx · 2025-07-02

**Rating:** 5
**Confidence:** 1

**Summary:**

Bubbleformer is a transformer-based model designed to forecast chaotic boiling dynamics, including temperature, velocity, and bubble interface evolution, without requiring future simulation inputs during inference. It tackles limitations in prior machine learning boiling models by introducing: Axial attention and frequency-aware scaling to capture spatiotemporal complexity, FiLM-based conditioning on physical parameters for cross-fluid and regime generalization, and BubbleML 2.0 which is a large-scale dataset with 160+ high-fidelity simulations across various boiling scenarios. Bubbleformer surpasses baseline models (e.g., UNet, FFNO) while generating physically plausible outputs.

**Dataset Code Accessibility:**

Yes

**Dataset Code Comments:**

The author provided the dataset in the public source.

**Ethical Comments:**

Not involved.

**Ethical Considerations:**

No, there are no or only very minor ethics concerns

**Final Justification:**

Satisfied with the rebuttal and rasie to 5.

**Limitations Weaknesses:**

1. Limited Physics Coverage\
Model is trained separately on subcooled and saturated conditions; cannot yet handle mixed physical regimes simultaneously.
2. Computational Cost\
Transformer architecture may still be resource-intensive compared to simpler models for specific tasks. The author should provide the cost.

**Strengths Contributions:**

1. Architectural Innovations.\
Axial attention reduces computational complexity and captures directional features in flow boiling, and frequency-aware attention mitigates oversmoothing, preserving sharp interfaces.
2. Superior Performance.\
Bubbleformer outperforms UNet and FFNO ,showcasihg the superiority.
3. A dataset expansion.\
BubbleML 2.0 offers the most extensive boiling simulation dataset to date.

---

> ### Author Rebuttal · Authors · 2025-07-31
>
> We thank the reviewer for their assessment and address the key concerns below.
>
> **Computational efficiency analysis.**
> Contrary to typical transformer concerns, we observe that Bubbleformer is efficient compared to the baseline models in the paper (UNet and FFNO). We report below the number of parameters and inference time per rollout timestep on a single NVIDIA A30 GPU.
>
> | Model            | Parameters (M) | Time per rollout step (s) |
> |:-----------------|---------------:|---------------------------:|
> | Bubbleformer-S   |           29.5 |                      0.016 |
> | Bubbleformer-L   |          115.8 |                      0.028 |
> | FFNO             |          119.3 |                      0.043 |
> | Modern UNet      |          566.7 |                      0.032 |
>
> The efficiency gain is mainly due to axial attention which reduces the complexity from $\mathcal{O}(HWT)^2$ in joint space-time attention to $\mathcal{O}(H^2 + W^2 + T^2)$. It is also important to note that the number of Fourier modes in the low pass filter in FFNO architectures largely determines its performance. Literature [1] dictates that the number of Fourier modes should be closer to two-thirds of the spatial domain resolution to preserve spectral information while avoiding aliasing errors. This effectively places a constraint on the number of parameters in the model to input resolution. The modern UNet architecture we chose is the one publicly released as part of BubbleML [2] benchmarks, and the effect on accuracy and inference time by reducing the number of parameters remains to be explored.
>
> **Unifying Physical Regimes.**
> Saturated boiling exhibits nucleate bubble formation with clear interface dynamics, while subcooled boiling involves complex thermal boundary layers and condensation effects. Unified architectures reduce performance due to these conflicting inductive biases and we are currently exploring the incorporation of physics-based losses and architectural improvements to improve the generalization of our models.
>
> [1] Incremental Spatial and Spectral Learning of Neural Operators for Solving Large-Scale PDEs, Robert Joseph George and Jiawei Zhao and Jean Kossaifi and Zongyi Li and Anima Anandkumar, TMLR 2024.
>
> [2] BubbleML: A Multiphase Multiphysics Dataset and Benchmarks for Machine Learning, Hassan et al, NeurIPS 2023.

---

> > ### Comment · Reviewer_X3Qx · 2025-08-02
> > **Satisfied with the rebuttal**
> >
> > Thanks for the experiments, and the authors addressed my concerns. I will raise the score to 5.

---

### Official Review · Reviewer_8fgs · 2025-07-04

**Rating:** 5
**Confidence:** 3

**Summary:**

This paper introduces a novel transformer architecture and model called Bubleformer and a corresponding dataset BubbleML2.0, a benchmark dataset for evaluating the various boiling configurations, fluid types and flow regimes.
The authors develop a new model which overcomes various limitations related with previous approaches through axial-attention and physics-informed architectural changes. The model developed therein is also shown to generalize across fluids, geometries, operating conditions and regimes. The model beats existing benchmarks in various boiling-flow related tasks.

**Additional Feedback:**

*Presentations/Grammar/Typos (Some might have been missed, will update if found later)*

Line 165: Mentions a 7D fluid descriptor, whereas the Appendix C.1 mentions 9-dimensional thermophysical input. I suppose it should be 9D at both places.

Line 883 Appendix B.1 The link to the model zoo appears broken to me and the model zoo directory does not feature in the repository structure outlined in the README.

Line 1073: ‘All’ instead of ‘Alll’

**Dataset Code Accessibility:**

Yes

**Dataset Code Comments:**

The entire code for the dataset evaluation is provided in a public github repository. The dataset is well structured, documented and is also made available through huggingface.

**Ethical Considerations:**

No, there are no or only very minor ethics concerns

**Final Justification:**

I had a query regarding unifying physical regimes which the authors have satisfactorily addressed in the rebuttal to reviewer X3Qx. The authors have also responded to my query, with experimental observations. The issue of non native support for AMR grids is not directly addressed as part of this work and is an important research direction. Considering all these points I would retain my score.

**Limitations Weaknesses:**

The authors have included a limitations section and have highlighted that the model is not suited for adaptive mesh refinement (AMR) grids.

A limitation of this work is the requirement of different models for Saturated and Subcooled pool boiling.

For the subcooled pool boiling task for Bubbleformer-S, out-of-distribution gives worse results (Appendix C.3), although Bubbleformer-L gives better results. So would this scenario and in general the model would benefit from an additional physics-informed loss function?
Was this considered or not, what was the rationale for the same? Can the equations in Appendix A be incorporated in the training? (This might improve out-of-distribution generalization of the model)

**Strengths Contributions:**

The authors provide an elaborate discussion of the existing landscape of the work in the introduction. The related works section provides enough information for someone who might not be very familiar and places itself well among the existing literature.

A strong contribution is the dataset BubbleML 2.0 which improves over BubbleML 1.0 by incorporating different types of fluids which exhibit different physics, different properties and covering various regimes as BubbleML 1.0 dataset includes only a dielectric fluid.

New, interpretable physics-based metrics have been introduced to evaluate the physical plausibility of the system.

Extremely detailed descriptions on experimental configurations, setup, model hyperparameters and ablations have been provided in the appendices.

---

> ### Author Rebuttal · Authors · 2025-07-31
>
> We thank the reviewer for the positive assessment and constructive feedback. We address the concerns below.
>
> **Physics-informed loss.** The reviewer raises an excellent point about incorporating physics-based losses to improve out-of-distribution generalization. We explored this direction with two approaches:
>
> 1. **Eikonal loss for interface tracking:** We incorporated the Eikonal equation (Equation 1) to enforce valid signed distance function properties for bubble interfaces. However, the results were worse than for the data-only optimization problem despite hyperparameter tuning.
>
> 2. **Continuity equation:** We added mass conservation constraints (Equation 5), using mass flux fields from our dataset. However, the high sparsity of the mass flux field created gradient instabilities and the models did not converge.
>
> The fundamental challenge is that boiling involves rapid phase transitions, creating discontinuous physics that standard PINN formulations struggle with. Moreover, adding multiple governing physical equations results in a highly non-linear loss landscape that is difficult to converge.
>
> We agree that developing physics-aware loss functions for discontinuous multiphase systems represents an important research direction that could enable true model unification and out-of-distribution generalization. We aim to continue pushing this boundary in future work.
>
> **Corrections and Typos**
>
> Line 165: Corrected to 9D fluid descriptor (matching Appendix C.1).
>
> Line 883: Model zoo link updated and verified. The [Github link](https://github.com/HPCForge/bubbleformer/tree/main/model-zoo) mentioned in the paper contains a redirection to the [Hugging Face](https://huggingface.co/hpcforge/Bubbleformer) repository containing the model weights
>
> Line 1073: Fixed typo "Alll" to "All".

---

### Note · Authors · 2025-08-13

We thank the reviewers for their constructive reviews and we are glad that all the reviewers have a positive assessment of our paper.

As our final comment we would like to address the final question by reviewer if6d. We apologize for the earlier confusion and would like to provide a clearer explanation on the current limitation of Bubbleformer models to operate on AMR grids.. Training on AMR data would require extending Bubbleformer to non-uniform grids or bring the AMR grids to a uniform resolution while enforcing divergence-free interpolation and flux conservation between fine and coarse levels. Incorporating these capabilities forms the basis of our planned future work.

At present, when we interpolate AMR simulation data to a uniform grid using simple bilinear or bicubic interpolation, it does not enforce the divergence-free criterion required for incompressibility. In contrast to simulations, which achieve divergence on the order of $10^{-12}$- $10^{-14}$, this approach is limited to approximately $10^{-2}$, introducing a 1–5% continuity error especially at the liquid-vapor interface. Thus the Navier-Stokes continuity equation is not satisfied to machine precision at the liquid-vapor interface when using standard interpolation techniques.

This issue is further compounded by a mismatch in grid layouts between the raw simulation and the training data. The numerical simulations employ a staggered mesh to maintain the divergence-free condition, whereas the training data contain velocities at the cell centers, which also contributes to the divergence errors observed.

Given that the primary focus of this paper was to demonstrate bubbleformer's ability to predict re-nucleation events and remain stable over long time periods, we have deferred this limitation to future work.

---

### Decision · Program_Chairs · 2025-09-18

**Decision:**

Accept (spotlight)

**Comment:**

This submission introduces Bubbleformer, a transformer-based spatiotemporal model that autonomously forecasts two-phase boiling—including nucleation and interface evolution—while conditioning on thermophysical parameters to generalize across fluids, geometries, and regimes; it is paired with BubbleML 2.0, a substantially broadened benchmark and a suite of physics-grounded evaluation metrics, with data and code openly released for reproducibility.  Empirically, the model establishes clear state-of-the-art results on prediction and forecasting tasks and remains stable over long horizons, with analyses showing physically plausible rollouts and improved long-range behavior versus prior UNet/FFNO baselines. The paper is clearly written and thorough (architectural details, ablations, and physically interpretable metrics), and the dataset contribution is likely to catalyze follow-on work in scientific ML for multiphase transport. The primary limitation is the lack of native support for AMR grids; in the discussion the authors clarified why naïve interpolation violates divergence-free constraints, outlined directions for AMR-native architectures, and noted that physics-aware losses (Eikonal and mass conservation) were explored but proved unstable in this discontinuous regime—useful transparency that frames concrete future work.  Reviewer–author discussion converged positively: concerns about DB-track fit and technical limitations were addressed, multiple reviewers acknowledged the rebuttal and raised their evaluations, and the final comments emphasized that AMR support is the only substantive remaining weakness.  Overall, the combination of a strong new benchmark, a compelling baseline model, rigorous physically based evaluation, and open resources promises high impact beyond this subarea, warranting a Spotlight.